# The Landscape of Non-convex Empirical Risk with Degenerate Population Risk

**Shuang Li, Gongguo Tang, and Michael B. Wakin**
Department of Electrical Engineering
Colorado School of Mines
Golden, CO 80401
{shuangli,gtang,mwakin}@mines.edu

## Abstract

The landscape of empirical risk has been widely studied in a series of machine learning problems, including low-rank matrix factorization, matrix sensing, matrix completion, and phase retrieval. In this work, we focus on the situation where the corresponding population risk is a degenerate non-convex loss function, namely, the Hessian of the population risk can have zero eigenvalues. Instead of analyzing the non-convex empirical risk directly, we first study the landscape of the corresponding population risk, which is usually easier to characterize, and then build a connection between the landscape of the empirical risk and its population risk. In particular, we establish a correspondence between the critical points of the empirical risk and its population risk without the strongly Morse assumption, which is required in existing literature but not satisfied in degenerate scenarios. We also apply the theory to matrix sensing and phase retrieval to demonstrate how to infer the landscape of empirical risk from that of the corresponding population risk.

## 1 Introduction

Understanding the connection between empirical risk and population risk can yield valuable insight into an optimization problem [1, 2]. Mathematically, the empirical risk $f(\boldsymbol{x})$ with respect to a parameter vector $\boldsymbol{x}$ is defined as

$$f(\boldsymbol{x}) \triangleq \frac{1}{M} \sum_{m=1}^{M} \mathcal{L}(\boldsymbol{x}, \boldsymbol{y}_m).$$

Here, $\mathcal{L}(\cdot)$ is a loss function and we are interested in losses that are non-convex in $\boldsymbol{x}$ in this work. $\boldsymbol{y} = [\boldsymbol{y}_1, \cdots, \boldsymbol{y}_M]^\top$ is a vector containing the random training samples, and $M$ is the total number of samples contained in the training set. The population risk, denoted as $g(\boldsymbol{x})$, is the expectation of the empirical risk with respect to the random measure used to generate the samples $\boldsymbol{y}$, i.e., $g(\boldsymbol{x}) = \mathbb{E}f(\boldsymbol{x})$.

Recently, the landscapes of empirical and population risk have been extensively studied in many fields of science and engineering, including machine learning and signal processing. In particular, the local or global geometry has been characterized in a wide variety of convex and non-convex problems, such as matrix sensing [3, 4], matrix completion [5, 6, 7], low-rank matrix factorization [8, 9, 10], phase retrieval [11, 12], blind deconvolution [13, 14], tensor decomposition [15, 16, 17], and so on. In this work, we focus on analyzing global geometry, which requires understanding not only regions near critical points but also the landscape away from these points.

It follows from empirical process theory that the empirical risk can uniformly converge to the corresponding population risk as $M \to \infty$ [18]. A recent work [1] exploits the uniform convergence of the empirical risk to the corresponding population risk and establishes a correspondence of their

critical points when provided with enough samples. The authors build their theoretical guarantees based on the assumption that the population risk is *strongly Morse*, namely, the Hessian of the population risk cannot have zero eigenvalues at or near the critical points[1]. However, many problems of practical interest do have Hessians with zero eigenvalues at some critical points. We refer to such problems as *degenerate*. To illustrate this, we present the very simple rank-1 matrix sensing and phase retrieval examples below.

**Example 1.1.** *(Rank-1 matrix sensing). Given measurements* $\boldsymbol{y}_m = \langle \mathbf{A}_m, \boldsymbol{x}^\star \boldsymbol{x}^{\star\top} \rangle, \ 1 \leq m \leq M,$ *where* $\boldsymbol{x}^\star \in \mathbb{R}^N$ *and* $\mathbf{A}_m \in \mathbb{R}^{N \times N}$ *denote the true signal and the* $m$*-th Gaussian sensing matrix with entries following* $\mathcal{N}(0, 1)$, *respectively. The following empirical risk is commonly used in practice*

$$f(\boldsymbol{x}) = \frac{1}{4M} \sum_{m=1}^M \left( \langle \mathbf{A}_m, \boldsymbol{x}\boldsymbol{x}^\top \rangle - \boldsymbol{y}_m \right)^2.$$

*The corresponding population risk is then*

$$g(\boldsymbol{x}) = \mathbb{E}f(\boldsymbol{x}) = \frac{1}{4}\|\boldsymbol{x}\boldsymbol{x}^\top - \boldsymbol{x}^\star \boldsymbol{x}^{\star\top}\|_F^2.$$

*Elementary calculations give the gradient and Hessian of the above population risk as*

$$\nabla g(\boldsymbol{x}) = (\boldsymbol{x}\boldsymbol{x}^\top - \boldsymbol{x}^\star \boldsymbol{x}^{\star\top})\boldsymbol{x}, \quad and \quad \nabla^2 g(\boldsymbol{x}) = 2\boldsymbol{x}\boldsymbol{x}^\top - \boldsymbol{x}^\star \boldsymbol{x}^{\star\top} + \|\boldsymbol{x}\|_2^2\mathbf{I}_N.$$

*We see that* $g(\boldsymbol{x})$ *has three critical points* $\boldsymbol{x} = \boldsymbol{0}, \ \pm \boldsymbol{x}^\star$. *Observe that the Hessian at* $\boldsymbol{x} = \boldsymbol{0}$ *is* $\nabla^2 g(\boldsymbol{0}) = -\boldsymbol{x}^\star \boldsymbol{x}^{\star\top}$, *which does have zero eigenvalues and thus* $g(\boldsymbol{x})$ *does not satisfy the strongly Morse condition required in [1]. The conclusion extends to the general low-rank matrix sensing.*

**Example 1.2.** *(Phase retrieval). Given measurements* $\boldsymbol{y}_m = |\langle \boldsymbol{a}_m, \boldsymbol{x}^\star \rangle|^2, \ 1 \leq m \leq M,$ *where* $\boldsymbol{x}^\star \in \mathbb{R}^N$ *and* $\boldsymbol{a}_m \in \mathbb{R}^N$ *denote the true signal and the* $m$*-th Gaussian random vector with entries following* $\mathcal{N}(0, 1)$, *respectively. The following empirical risk is commonly used in practice*

$$f(\boldsymbol{x}) = \frac{1}{2M} \sum_{m=1}^M \left( |\langle \boldsymbol{a}_m, \boldsymbol{x} \rangle|^2 - \boldsymbol{y}_m \right)^2. \tag{1.1}$$

*The corresponding population risk is then*

$$g(\boldsymbol{x}) = \mathbb{E}f(\boldsymbol{x}) = \|\boldsymbol{x}\boldsymbol{x}^\top - \boldsymbol{x}^\star \boldsymbol{x}^{\star\top}\|_F^2 + \frac{1}{2}(\|\boldsymbol{x}\|_2^2 - \|\boldsymbol{x}^\star\|_2^2)^2. \tag{1.2}$$

*Elementary calculations give the gradient and Hessian of the above population risk as*

$$\nabla g(\boldsymbol{x}) = 6\|\boldsymbol{x}\|_2^2\boldsymbol{x} - 2\|\boldsymbol{x}^\star\|_2^2\boldsymbol{x} - 4(\boldsymbol{x}^{\star\top}\boldsymbol{x})\boldsymbol{x}^\star,$$
$$\nabla^2 g(\boldsymbol{x}) = 12\boldsymbol{x}\boldsymbol{x}^\top - 4\boldsymbol{x}^\star \boldsymbol{x}^{\star\top} + 6\|\boldsymbol{x}\|_2^2\mathbf{I}_N - 2\|\boldsymbol{x}^\star\|_2^2\mathbf{I}_N.$$

*We see that the population loss has critical points* $\boldsymbol{x} = \boldsymbol{0}, \ \pm \boldsymbol{x}^\star, \ \frac{1}{\sqrt{3}}\|\boldsymbol{x}^\star\|_2\boldsymbol{w}$ *with* $\boldsymbol{w}^\top \boldsymbol{x}^\star = 0$ *and* $\|\boldsymbol{w}\|_2 = 1$. *Observe that the Hessian at* $\boldsymbol{x} = \frac{1}{\sqrt{3}}\|\boldsymbol{x}^\star\|_2\boldsymbol{w}$ *is* $\nabla^2 g(\frac{1}{\sqrt{3}}\|\boldsymbol{x}^\star\|_2\boldsymbol{w}) = 4\|\boldsymbol{x}^\star\|_2^2\boldsymbol{w}\boldsymbol{w}^\top - 4\boldsymbol{x}^\star \boldsymbol{x}^{\star\top}$, *which also has zero eigenvalues and thus* $g(\boldsymbol{x})$ *does not satisfy the strongly Morse condition required in [1].*

In this work, we aim to fill this gap and establish the correspondence between the critical points of empirical risk and its population risk *without* the strongly Morse assumption. In particular, we work on the situation where the population risk may be a *degenerate* non-convex function, i.e., the Hessian of the population risk can have zero eigenvalues. Given the correspondence between the critical points of the empirical risk and its population risk, we are able to build a connection between the landscape of the empirical risk and its population counterpart. To illustrate the effectiveness of this theory, we also apply it to applications such as matrix sensing (with general rank) and phase retrieval to show how to characterize the landscape of the empirical risk via its corresponding population risk.

The remainder of this work is organized as follows. In Section 2, we present our main results on the correspondence between the critical points of the empirical risk and its population risk. In Section 3, we apply our theory to the two applications, matrix sensing and phase retrieval. In Section 4, we conduct experiments to further support our analysis. Finally, we conclude our work in Section 5.

**Notation:** For a twice differential function $f(\cdot)$: $\nabla f$, $\nabla^2 f$, $\mathrm{grad}\, f$, and $\mathrm{hess}\, f$ denote the gradient and Hessian of $f$ in the Euclidean space and with respect to a Riemannian manifold $\mathcal{M}$, respectively. Note that the Riemannian gradient/Hessian (grad/hess) reduces to the Euclidean gradient/Hessian ($\nabla/\nabla^2$) when the domain of $f$ is the Euclidean space. For a scalar function with a matrix variable, e.g., $f(\mathbf{U})$, we represent its Euclidean Hessian with a bilinear form defined as $\nabla^2 f(\mathbf{U})[\mathbf{D},\mathbf{D}] = \sum_{i,j,p,q}\frac{\partial^2 f(\mathbf{U})}{\partial\mathbf{D}(i,j)\partial\mathbf{D}(p,q)}\mathbf{D}(i,j)\mathbf{D}(p,q)$ for any $\mathbf{D}$ having the same size as $\mathbf{U}$. Denote $\mathcal{B}(l)$ as a compact and connected subset of a Riemannian manifold $\mathcal{M}$ with $l$ being a problem-specific parameter.[2]

## 2 Main Results

In this section, we present our main results on the correspondence between the critical points of the empirical risk and its population risk. Let $\mathcal{M}$ be a Riemannian manifold. For notational simplicity, we use $\boldsymbol{x} \in \mathcal{M}$ to denote the parameter vector when we introduce our theory[3]. We begin by introducing the assumptions needed to build our theory. Denote $f(\boldsymbol{x})$ and $g(\boldsymbol{x})$ as the empirical risk and the corresponding population risk defined for $\boldsymbol{x} \in \mathcal{M}$, respectively. Let $\epsilon$ and $\eta$ be two positive constants.

**Assumption 2.1.** *The population risk $g(\boldsymbol{x})$ satisfies*

$$|\lambda_{\min}(\mathrm{hess}\, g(\boldsymbol{x}))| \geq \eta \qquad (2.1)$$

*in the set $\overline{\mathcal{D}} \triangleq \{\boldsymbol{x} \in \mathcal{B}(l) : \|\mathrm{grad}\, g(\boldsymbol{x})\|_2 \leq \epsilon\}$. Here, $\lambda_{\min}(\cdot)$ denotes the minimal eigenvalue (not the eigenvalue of smallest magnitude).*

Assumption 2.1 is closely related to the robust strict saddle property [19] – it requires that any point with a small gradient has either a positive definite Hessian ($\lambda_{\min}(\mathrm{hess}\, g(\boldsymbol{x})) \geq \eta$) or a Hessian with a negative curvature ($\lambda_{\min}(\mathrm{hess}\, g(\boldsymbol{x})) \leq -\eta$). It is weaker than the $(\epsilon, \eta)$-strongly Morse condition as it allows the Hessian $\mathrm{hess}\, g(\boldsymbol{x})$ to have zero eigenvalues in $\overline{\mathcal{D}}$, provided it also has at least one sufficiently negative eigenvalue.

**Assumption 2.2.** *(Gradient proximity). The gradients of the empirical risk and population risk satisfy*

$$\sup_{\boldsymbol{x}\in\mathcal{B}(l)} \|\mathrm{grad}\, f(\boldsymbol{x}) - \mathrm{grad}\, g(\boldsymbol{x})\|_2 \leq \frac{\epsilon}{2}. \qquad (2.2)$$

**Assumption 2.3.** *(Hessian proximity). The Hessians of the empirical risk and population risk satisfy*

$$\sup_{\boldsymbol{x}\in\mathcal{B}(l)} \|\mathrm{hess}\, f(\boldsymbol{x}) - \mathrm{hess}\, g(\boldsymbol{x})\|_2 \leq \frac{\eta}{2}. \qquad (2.3)$$

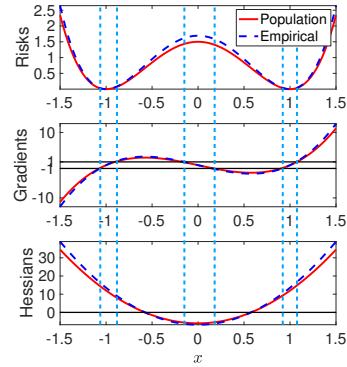

Figure 1: Phase retrieval with $N = 1$.

To illustrate the above three assumptions, we use the phase retrieval Example 1.2 with $N = 1$, $\boldsymbol{x}^\star = 1$, and $M = 30$. We present the population risk $g(x) = \frac{3}{2}(x^2-1)^2$ and the empirical risk $f(x) = \frac{1}{2M}\sum_{m=1}^{M} a_m^4 (x^2-1)^2$ together with their gradients and Hessians in Figure 1. It can be seen that in the small gradient region (the three parts between the light blue vertical dashed lines), the absolute value of the population Hessian's minimal eigenvalue (which equals the absolute value of Hessian here since $N = 1$) is bounded away from zero. In addition, with enough measurements, e.g., $M = 30$, we do see the gradients and Hessians of the empirical and population risk are close to each other.

We are now in the position to state our main theorem.

**Theorem 2.1.** *Denote $f$ and $g$ as the non-convex empirical risk and the corresponding population risk, respectively. Let $\mathcal{D}$ be any maximal connected and compact subset of $\overline{\mathcal{D}}$ with a $\mathcal{C}^2$ boundary $\partial\mathcal{D}$. Under Assumptions 2.1-2.3 stated above, the following statements hold:*

**(a)** $\mathcal{D}$ *contains at most one local minimum of* $g$*. If* $g$ *has* $K$ $(K = 0, 1)$ *local minima in* $\mathcal{D}$*, then* $f$ *also has* $K$ *local minima in* $\mathcal{D}$*.*

**(b)** *If* $g$ *has strict saddles in* $\mathcal{D}$*, then if* $f$ *has any critical points in* $\mathcal{D}$*, they must be strict saddle points.*

The proof of Theorem 2.1 is given in Appendix A (see supplementary material). In particular, we prove Theorem 2.1 by extending the proof of Theorem 2 in [1] *without* requiring the strongly Morse assumption on the population risk. We first present two key lemmas, in which we show that there exists a correspondence between the critical points of the empirical risk and those of the population risk in a connected and compact set under certain assumptions, and the small gradient area can be partitioned into many maximal connected and compact components with each component either containing one local minimum or no local minimum. Finally, we finish the proof of Theorem 2.1 by using these two key lemmas.

Part (a) in Theorem 2.1 indicates a one-to-one correspondence between the local minima of the empirical risk and its population risk. We can further bound the distance between the local minima of the empirical risk and its population risk. We summarize this result in the following corollary, which is proved in Appendix C (see supplementary material).

**Corollary 2.1.** *Let* $\{\widehat{\boldsymbol{x}}_k\}_{k=1}^K$ *and* $\{\boldsymbol{x}_k\}_{k=1}^K$ *denote the local minima of the empirical risk and its population risk, and* $\mathcal{D}_k$ *be the maximal connected and compact subset of* $\overline{\mathcal{D}}$ *containing* $\boldsymbol{x}_k$ *and* $\widehat{\boldsymbol{x}}_k$*. Let* $\rho$ *be the* injectivity radius *of the manifold* $\mathcal{M}$*. Suppose the pre-image of* $\mathcal{D}_k$ *under the exponential mapping* $\mathrm{Exp}_{\boldsymbol{x}_k}(\cdot)$ *is contained in the ball at the origin of the tangent space* $\mathcal{T}_{\boldsymbol{x}_k}\mathcal{M}$ *with radius* $\rho$*. Assume the differential of the exponential mapping* $\mathrm{DExp}_{\boldsymbol{x}_k}(\boldsymbol{v})$ *has an operator norm bounded by* $\sigma$ *for all* $\boldsymbol{v} \in \mathcal{T}_{\boldsymbol{x}_k}\mathcal{M}$ *with norm less than* $\rho$*. Suppose the pullback of the population risk onto the tangent space* $\mathcal{T}_{\boldsymbol{x}_k}\mathcal{M}$ *has Lipschitz Hessian with constant* $L_H$ *at the origin. Then as long as* $\epsilon \leq \frac{\eta^2}{2\sigma L_H}$*, the Riemannian distance between* $\widehat{\boldsymbol{x}}_k$ *and* $\boldsymbol{x}_k$ *satisfies*

$$dist(\widehat{\boldsymbol{x}}_k, \boldsymbol{x}_k) \leq 2\sigma\epsilon/\eta, \quad 1 \leq k \leq K.$$

In general, the two parameters $\epsilon$ and $\eta$ used in Assumptions 2.1-2.3 can be obtained by lower bounding $|\lambda_{\min}(\mathrm{hess}\, g(\boldsymbol{x}))|$ in a small gradient region. In this way, one can adjust the size of the small gradient region to get an upper bound on $\epsilon$, and use the lower bound for $|\lambda_{\min}(\mathrm{hess}\, g(\boldsymbol{x}))|$ as $\eta$. In the case when it is not easy to directly bound $|\lambda_{\min}(\mathrm{hess}\, g(\boldsymbol{x}))|$ in a small gradient region, one can also first choose a region for which it is easy to find the lower bound, and then show that the gradient has a large norm outside of this region, as we do in Section 3. For phase retrieval, note that $|\lambda_{\min}(\nabla^2 g(\boldsymbol{x}))|$ and $\|\nabla g(\boldsymbol{x})\|_2$ roughly scale with $\|\boldsymbol{x}^\star\|_2^2$ and $\|\boldsymbol{x}^\star\|_2^3$ in the regions near critical points, which implies that $\eta$ and the upper bound on $\epsilon$ should also scale with $\|\boldsymbol{x}^\star\|_2^2$ and $\|\boldsymbol{x}^\star\|_2^3$, respectively. For matrix sensing, in a similar way, $|\lambda_{\min}(\mathrm{hess}\, g(\mathbf{U}))|$ and $\|\mathrm{grad}\, g(\mathbf{U})\|_F$ roughly scale with $\lambda_k$ and $\lambda_k^{1.5}$ in the regions near critical points, which implies that $\eta$ and the upper bound on $\epsilon$ should also scale with $\lambda_k$ and $\lambda_k^{1.5}$, respectively. Note, however, with more samples (larger $M$), $\epsilon$ can be set to smaller values, while $\eta$ typically remains unchanged. One can refer to Section 3 for more details on the notation as well as how to choose $\eta$ and upper bounds on $\epsilon$ in the two applications.

Note that we have shown the correspondence between the critical points of the empirical risk and its population risk *without* the strongly Morse assumption in the above theorem. In particular, we relax the strongly Morse assumption to our Assumption 2.1, which implies that we are able to handle the scenario where the Hessian of the population risk has zero eigenvalues at some critical points or even everywhere in the set $\overline{\mathcal{D}}$. With this correspondence, we can then establish a connection between the landscape of the empirical risk and the population risk, and thus for problems where the population risk has a favorable geometry, we are able to carry this favorable geometry over to the corresponding empirical risk. To illustrate this in detail, we highlight two applications, matrix sensing and phase retrieval, in the next section.

## 3 Applications

In this section, we illustrate how to completely characterize the landscape of an empirical risk from its population risk using Theorem 2.1. In particular, we apply Theorem 2.1 to two applications, matrix sensing and phase retrieval. In order to use Theorem 2.1, all we need is to verify that the empirical risk and population risk in these two applications satisfy the three assumptions stated in Section 2.

## 3.1 Matrix Sensing

Let $\mathbf{X} \in \mathbb{R}^{N \times N}$ be a symmetric, positive semi-definite matrix with rank $r$. We measure $\mathbf{X}$ with a symmetric Gaussian linear operator $\mathcal{A} : \mathbb{R}^{N \times N} \to \mathbb{R}^M$. The $m$-th entry of the observation $\boldsymbol{y} = \mathcal{A}(\mathbf{X})$ is given as $\boldsymbol{y}_m = \langle \mathbf{X}, \mathbf{A}_m \rangle$, where $\mathbf{A}_m = \frac{1}{2}(\mathbf{B}_m + \mathbf{B}_m^\top)$ with $\mathbf{B}_m$ being a Gaussian random matrix with entries following $\mathcal{N}(0, \frac{1}{M})$. The adjoint operator $\mathcal{A}^* : \mathbb{R}^M \to \mathbb{R}^{N \times N}$ is defined as $\mathcal{A}^*(\boldsymbol{y}) = \sum_{m=1}^{M} \boldsymbol{y}_m \mathbf{A}_m$. It can be shown that $\mathbb{E}(\mathcal{A}^* \mathcal{A})$ is the identity operator, i.e. $\mathbb{E}(\mathcal{A}^* \mathcal{A}(\mathbf{X})) = \mathbf{X}$. To find a low-rank approximation of $\mathbf{X}$ when given the measurements $\boldsymbol{y} = \mathcal{A}(\mathbf{X})$, one can solve the following optimization problem:

$$\min_{\widetilde{\mathbf{X}} \in \mathbb{R}^{N \times N}} \frac{1}{4} \|\mathcal{A}(\widetilde{\mathbf{X}} - \mathbf{X})\|_2^2 \quad \text{s.t.} \quad \text{rank}(\widetilde{\mathbf{X}}) \le k, \widetilde{\mathbf{X}} \succeq 0. \tag{3.1}$$

Here, we assume that $\frac{r}{2} \le k \le r \ll N$. By using the Burer-Monteiro type factorization [20, 21], i.e., letting $\widetilde{\mathbf{X}} = \mathbf{U}\mathbf{U}^\top$ with $\mathbf{U} \in \mathbb{R}^{N \times k}$, we can transform the above optimization problem into the following unconstrained one:

$$\min_{\mathbf{U} \in \mathbb{R}^{N \times k}} f(\mathbf{U}) \triangleq \frac{1}{4} \|\mathcal{A}(\mathbf{U}\mathbf{U}^\top - \mathbf{X})\|_2^2. \tag{3.2}$$

Observe that this empirical risk $f(\mathbf{U})$ is a non-convex function due to the quadratic term $\mathbf{U}\mathbf{U}^\top$. With some elementary calculation, we obtain the gradient and Hessian of $f(\mathbf{U})$, which are given as

$$\nabla f(\mathbf{U}) = \mathcal{A}^* \mathcal{A}(\mathbf{U}\mathbf{U}^\top - \mathbf{X})\mathbf{U},$$

$$\nabla^2 f(\mathbf{U})[\mathbf{D}, \mathbf{D}] = \frac{1}{2} \|\mathcal{A}(\mathbf{U}\mathbf{D}^\top + \mathbf{D}\mathbf{U}^\top)\|_2^2 + \langle \mathcal{A}^* \mathcal{A}(\mathbf{U}\mathbf{U}^\top - \mathbf{X}), \mathbf{D}\mathbf{D}^\top \rangle.$$

Computing the expectation of $f(\mathbf{U})$, we get the population risk

$$g(\mathbf{U}) = \mathbb{E}f(\mathbf{U}) = \frac{1}{4} \|\mathbf{U}\mathbf{U}^\top - \mathbf{X}\|_F^2, \tag{3.3}$$

whose gradient and Hessian are given as

$$\nabla g(\mathbf{U}) = (\mathbf{U}\mathbf{U}^\top - \mathbf{X})\mathbf{U} \quad \text{and} \quad \nabla^2 g(\mathbf{U})[\mathbf{D}, \mathbf{D}] = \frac{1}{2} \|\mathbf{U}\mathbf{D}^\top + \mathbf{D}\mathbf{U}^\top\|_2^2 + \langle \mathbf{U}\mathbf{U}^\top - \mathbf{X}, \mathbf{D}\mathbf{D}^\top \rangle.$$

The landscape of the above population risk has been studied in the general $\mathbb{R}^{N \times k}$ space with $k = r$ in [8]. The landscape of its variants, such as the asymmetric version with or without a balanced term, has also been studied in [4, 22]. It is well known that there exists an ambiguity in the solution of (3.2) due to the fact that $\mathbf{U}\mathbf{U}^\top = \mathbf{U}\mathbf{Q}\mathbf{Q}^\top\mathbf{U}^\top$ holds for any orthogonal matrix $\mathbf{Q} \in \mathbb{R}^{k \times k}$. This implies that the Euclidean Hessian $\nabla^2 g(\mathbf{U})$ always has zero eigenvalues for $k > 1$ at critical points, even at local minima, violating not only the strongly Morse condition but also Assumption 2.1. To overcome this difficulty, we propose to formulate an equivalent problem on a proper quotient manifold (rather than the general $\mathbb{R}^{N \times k}$ space as in [8]) to remove this ambiguity and make sure Assumption 2.1 is satisfied.

### 3.1.1 Background on the quotient manifold

To keep our work self-contained, we provide a brief introduction to quotient manifolds in this section before we verify our three assumptions. One can refer to [23, 24] for more information. We make the assumption that the matrix variable $\mathbf{U}$ is always full-rank. This is required in order to define a proper quotient manifold, since otherwise the equivalence classes defined below will have different dimensions, violating Proposition 3.4.4 in [23]. Thus, we focus on the case that $\mathbf{U}$ belongs to the manifold $\mathbb{R}_*^{N \times k}$, i.e., the set of all $N \times k$ real matrices with full column rank. To remove the parameterization ambiguity caused by the factorization $\widetilde{\mathbf{X}} = \mathbf{U}\mathbf{U}^\top$, we define an *equivalence class* for any $\mathbf{U} \in \mathbb{R}_*^{N \times k}$ as $[\mathbf{U}] \triangleq \{\mathbf{V} \in \mathbb{R}_*^{N \times k} : \mathbf{V}\mathbf{V}^\top = \mathbf{U}\mathbf{U}^\top\} = \{\mathbf{U}\mathbf{Q} : \mathbf{Q} \in \mathbb{R}^{k \times k}, \mathbf{Q}^\top\mathbf{Q} = \mathbf{I}_k\}$. We will abuse notation and use $\mathbf{U}$ to denote also its equivalence class $[\mathbf{U}]$ in the following. Let $\mathcal{M}$ denote the set of all equivalence classes of the above form, which admits a (unique) differential structure that makes it a (Riemannian) *quotient manifold*, denoted as $\mathcal{M} = \mathbb{R}_*^{N \times k}/\mathcal{O}_k$. Here $\mathcal{O}_k$ is the orthogonal group $\{\mathbf{Q} \in \mathbb{R}^{k \times k} : \mathbf{Q}\mathbf{Q}^\top = \mathbf{Q}^\top\mathbf{Q} = \mathbf{I}_k\}$. Since the objective function $g(\mathbf{U})$ in

(3.3) (and $f(\mathbf{U})$ in (3.2)) is invariant under the equivalence relation, it induces a unique function on the quotient manifold $\mathbb{R}_*^{N\times k}/\mathcal{O}_k$, also denoted as $g(\mathbf{U})$.

Note that the tangent space $\mathcal{T}_{\mathbf{U}}\mathbb{R}_*^{N\times k}$ of the manifold $\mathbb{R}_*^{N\times k}$ at any point $\mathbf{U}\in\mathbb{R}_*^{N\times k}$ is still $\mathbb{R}_*^{N\times k}$. We define the *vertical space* $\mathcal{V}_{\mathbf{U}}\mathcal{M}$ as the tangent space to the equivalence classes (which are themselves manifolds): $\mathcal{V}_{\mathbf{U}}\mathcal{M} \triangleq \{\mathbf{U}\boldsymbol{\Omega} : \boldsymbol{\Omega}\in\mathbb{R}^{k\times k}, \boldsymbol{\Omega}^\top = -\boldsymbol{\Omega}\}$. We also define the *horizontal space* $\mathcal{H}_{\mathbf{U}}\mathcal{M}$ as the orthogonal complement of the vertical space $\mathcal{V}_{\mathbf{U}}\mathcal{M}$ in the tangent space $\mathcal{T}_{\mathbf{U}}\mathbb{R}_*^{N\times k} = \mathbb{R}_*^{N\times k}$: $\mathcal{H}_{\mathbf{U}}\mathcal{M} \triangleq \{\mathbf{D}\in\mathbb{R}_*^{N\times k} : \mathbf{D}^\top\mathbf{U} = \mathbf{U}^\top\mathbf{D}\}$. For any matrix $\mathbf{Z}\in\mathbb{R}_*^{N\times k}$, its projection onto the horizontal space $\mathcal{H}_{\mathbf{U}}\mathcal{M}$ is given as $\mathcal{P}_{\mathbf{U}}(\mathbf{Z}) = \mathbf{Z} - \mathbf{U}\boldsymbol{\Omega}$, where $\boldsymbol{\Omega}$ is a skew-symmetric matrix that solves the following Sylvester equation $\boldsymbol{\Omega}\mathbf{U}^\top\mathbf{U} + \mathbf{U}^\top\mathbf{U}\boldsymbol{\Omega} = \mathbf{U}^\top\mathbf{Z} - \mathbf{Z}^\top\mathbf{U}$. Then, we can define the Riemannian gradient (grad $\cdot$) and Hessian (hess $\cdot$) of the empirical risk and population risk on the quotient manifold $\mathcal{M}$, which are given in the supplementary material.

### 3.1.2 Verifying Assumptions 2.1, 2.2, and 2.3

Assume that $\mathbf{X} = \mathbf{W}\boldsymbol{\Lambda}\mathbf{W}^\top$ with $\mathbf{W}\in\mathbb{R}^{N\times r}$ and $\boldsymbol{\Lambda} = \mathrm{diag}([\lambda_1,\cdots,\lambda_r])\in\mathbb{R}^{r\times r}$ is an eigen-decomposition of $\mathbf{X}$. Without loss of generality, we assume that the eigenvalues of $\mathbf{X}$ are in descending order. Let $\boldsymbol{\Lambda}_u\in\mathbb{R}^{k\times k}$ be a diagonal matrix that contains any $k$ non-zero eigenvalues of $\mathbf{X}$ and $\mathbf{W}_u\in\mathbb{R}^{N\times k}$ contain the $k$ eigenvectors of $\mathbf{X}$ associated with the eigenvalues in $\boldsymbol{\Lambda}_u$. Let $\boldsymbol{\Lambda}_k = \mathrm{diag}([\lambda_1,\cdots,\lambda_k])$ be the diagonal matrix that contains the largest $k$ eigenvalues of $\mathbf{X}$ and $\mathbf{W}_k\in\mathbb{R}^{N\times k}$ contain the $k$ eigenvectors of $\mathbf{X}$ associated with the eigenvalues in $\boldsymbol{\Lambda}_k$. $\mathbf{Q}\in\mathcal{O}_k$ is any orthogonal matrix. The following lemma provides the global geometry of the population risk in (3.3), which also determines the values of $\epsilon$ and $\eta$ in Assumption 2.1.

**Lemma 3.1.** *Define* $\mathcal{U} \triangleq \{\mathbf{U} = \mathbf{W}_u\boldsymbol{\Lambda}_u^{\frac{1}{2}}\mathbf{Q}^\top\}$, $\mathcal{U}^\star \triangleq \{\mathbf{U}^\star = \mathbf{W}_k\boldsymbol{\Lambda}_k^{\frac{1}{2}}\mathbf{Q}^\top\} \subseteq \mathcal{U}$, *and* $\mathcal{U}_s^\star \triangleq \mathcal{U}\backslash\mathcal{U}^\star$. *Denote* $\kappa \triangleq \sqrt{\frac{\lambda_1}{\lambda_k}} \geq 1$ *as the condition number of any* $\mathbf{U}^\star\in\mathcal{U}^\star$. *Define the following regions:*

$$\mathcal{R}_1 \triangleq \left\{\mathbf{U}\in\mathbb{R}_*^{N\times k} : \min_{\mathbf{P}\in\mathcal{O}_k}\|\mathbf{U} - \mathbf{U}^\star\mathbf{P}\|_F \leq 0.2\kappa^{-1}\sqrt{\lambda_k}, \forall \mathbf{U}^\star\in\mathcal{U}^\star\right\},$$

$$\mathcal{R}_2' \triangleq \left\{\mathbf{U}\in\mathbb{R}_*^{N\times k} : \sigma_k(\mathbf{U}) \leq \frac{1}{2}\sqrt{\lambda_k}, \|\mathbf{U}\mathbf{U}^\top\|_F \leq \frac{8}{7}\|\mathbf{U}^\star\mathbf{U}^{\star\top}\|_F, \|\operatorname{grad} g(\mathbf{U})\|_F \leq \frac{1}{80}\lambda_k^{\frac{3}{2}}\right\},$$

$$\mathcal{R}_2'' \triangleq \left\{\mathbf{U}\in\mathbb{R}_*^{N\times k} : \sigma_k(\mathbf{U}) \leq \frac{1}{2}\sqrt{\lambda_k}, \|\mathbf{U}\mathbf{U}^\top\|_F \leq \frac{8}{7}\|\mathbf{U}^\star\mathbf{U}^{\star\top}\|_F, \|\operatorname{grad} g(\mathbf{U})\|_F > \frac{1}{80}\lambda_k^{\frac{3}{2}}\right\},$$

$$\mathcal{R}_3' \triangleq \left\{\mathbf{U}\in\mathbb{R}_*^{N\times k} : \sigma_k(\mathbf{U}) > \frac{1}{2}\sqrt{\lambda_k}, \min_{\mathbf{P}\in\mathcal{O}_k}\|\mathbf{U} - \mathbf{U}^\star\mathbf{P}\|_F > 0.2\kappa^{-1}\sqrt{\lambda_k}, \|\mathbf{U}\mathbf{U}^\top\|_F \leq \frac{8}{7}\|\mathbf{U}^\star\mathbf{U}^{\star\top}\|_F\right\},$$

$$\mathcal{R}_3'' \triangleq \left\{\mathbf{U}\in\mathbb{R}_*^{N\times k} : \|\mathbf{U}\mathbf{U}^\top\|_F > \frac{8}{7}\|\mathbf{U}^\star\mathbf{U}^{\star\top}\|_F\right\},$$

*where* $\sigma_k(\mathbf{U})$ *denotes the $k$-th singular value of a matrix* $\mathbf{U}\in\mathbb{R}_*^{N\times k}$, *i.e., the smallest singular value of* $\mathbf{U}$. *These regions also induce regions in the quotient manifold* $\mathcal{M}$ *in an apparent way. We additionally assume that* $\lambda_{k+1} \leq \frac{1}{12}\lambda_k$ *and* $k \leq r \ll N$. *Then, the following properties hold:*

*(1) For any* $\mathbf{U}\in\mathcal{U}$, $\mathbf{U}$ *is a critical point of the population risk* $g(\mathbf{U})$ *in (3.3).*

*(2) For any* $\mathbf{U}^\star\in\mathcal{U}^\star$, $\mathbf{U}^\star$ *is a global minimum of* $g(\mathbf{U})$ *with* $\lambda_{\min}(\operatorname{hess} g(\mathbf{U}^\star)) \geq 1.91\lambda_k$. *Moreover, for any* $\mathbf{U}\in\mathcal{R}_1$, *we have*

$$\lambda_{\min}(\operatorname{hess} g(\mathbf{U})) \geq 0.19\lambda_k.$$

*(3) For any* $\mathbf{U}_s^\star\in\mathcal{U}_s^\star$, $\mathbf{U}_s^\star$ *is a strict saddle point of* $g(\mathbf{U})$ *with* $\lambda_{\min}(\operatorname{hess} g(\mathbf{U}_s^\star)) \leq -0.91\lambda_k$. *Moreover, for any* $\mathbf{U}\in\mathcal{R}_2'$, *we have*

$$\lambda_{\min}(\operatorname{hess} g(\mathbf{U})) \leq -0.06\lambda_k.$$

*(4) For any* $\mathbf{U}\in\mathcal{R}_2''\bigcup\mathcal{R}_3'\bigcup\mathcal{R}_3''$, *we have a large gradient. In particular,*

$$\|\operatorname{grad} g(\mathbf{U})\|_F > \begin{cases} \frac{1}{80}\lambda_k^{\frac{3}{2}}, & \text{if } \mathbf{U}\in\mathcal{R}_2'', \\ \frac{1}{60}\kappa^{-1}\lambda_k^{\frac{3}{2}}, & \text{if } \mathbf{U}\in\mathcal{R}_3', \\ \frac{5}{84}k^{\frac{1}{4}}\lambda_k^{\frac{3}{2}}, & \text{if } \mathbf{U}\in\mathcal{R}_3''. \end{cases}$$

The proof of Lemma 3.1 is inspired by the proofs of [8, Theorem 4], [3, Lemma 13] and [4, Theorem 5], and is given in Appendix D (see supplementary material). Therefore, we can set $\epsilon \le \min\{1/80, 1/60\kappa^{-1}\}\lambda_k^{\frac{3}{2}}$ and $\eta = 0.06\lambda_k$. Then, the population risk given in (3.3) satisfies Assumption 2.1. It can be seen that each critical point of the population risk $g(\mathbf{U})$ in (3.3) is either a global minimum or a strict saddle, which inspires us to carry this favorable geometry over to the corresponding empirical risk.

To illustrate the partition of the manifold $\mathbb{R}_*^{N\times k}$ used in the above Lemma 3.1, we use the purple (①), yellow (②), and green (③) regions in Figure 2 to denote the regions that satisfy $\min_{\mathbf{P}\in\mathcal{O}_k}\|\mathbf{U} - \mathbf{U}^\star\mathbf{P}\|_F \le 0.2\kappa^{-1}\sqrt{\lambda_k}$, $\sigma_k(\mathbf{U}) \le \frac{1}{2}\sqrt{\lambda_k}$, and $\|\mathbf{U}\mathbf{U}^\top\|_F \le \frac{8}{7}\|\mathbf{U}^\star\mathbf{U}^{\star\top}\|_F$, respectively. It can be seen that $\mathcal{R}_1$ is exactly the purple region, which contains the areas near the global minima ($[\mathbf{U}^\star]$). $\mathcal{R}_2 = \mathcal{R}_2'\bigcup\mathcal{R}_2''$ is the intersection of the yellow and green regions. $\mathcal{R}_3'$ is the part of the green region that does not intersect with the purple or yellow regions. Finally, $\mathcal{R}_3''$ is the space outside of the green region. Therefore, the union of $\mathcal{R}_1$, $\mathcal{R}_2$, and $\mathcal{R}_3 = \mathcal{R}_3'\bigcup\mathcal{R}_3''$ covers the entire manifold $\mathbb{R}_*^{N\times k}$.

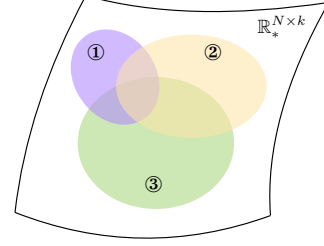

Figure 2: Partition of regions in Lemma 3.1.

We define a norm ball as $\mathcal{B}(l) \triangleq \{\mathbf{U} \in \mathbb{R}_*^{N\times k} : \|\mathbf{U}\mathbf{U}^\top\|_F \le l\}$ with $l = \frac{8}{7}\|\mathbf{U}^\star\mathbf{U}^{\star\top}\|_F$. The following lemma verifies Assumptions 2.2 and 2.3 under the restricted isometry property (RIP).

**Lemma 3.2.** *Assume $\frac{r}{2} \le k \le r \ll N$. Suppose that a linear operator $\mathcal{B}$ with $[\mathcal{B}(\mathbf{Z})]_m = \langle\mathbf{Z}, \mathbf{B}_m\rangle$ satisfies the following RIP*

$$(1 - \delta_{r+k})\|\mathbf{Z}\|_F^2 \le \|\mathcal{B}(\mathbf{Z})\|_2^2 \le (1 + \delta_{r+k})\|\mathbf{Z}\|_F^2 \tag{3.4}$$

*for any matrix $\mathbf{Z} \in \mathbb{R}^{N\times N}$ with rank at most $r + k$. We construct the linear operator $\mathcal{A}$ by setting $\mathbf{A}_m = \frac{1}{2}(\mathbf{B}_m + \mathbf{B}_m^\top)$. If the restricted isometry constant $\delta_{r+k}$ satisfies*

$$\delta_{r+k} \le \min\left\{\frac{\epsilon}{2\sqrt{\frac{8}{7}}k^{\frac{1}{4}}(\frac{8}{7}\|\mathbf{U}^\star\mathbf{U}^{\star\top}\|_F + \|\mathbf{X}\|_F)\|\mathbf{U}^\star\mathbf{U}^{\star\top}\|_F^{\frac{1}{2}}}, \frac{1}{36}, \frac{\eta}{2(\frac{16}{7}\sqrt{k}\|\mathbf{U}^\star\mathbf{U}^{\star\top}\|_F + \frac{8}{7}\|\mathbf{U}^\star\mathbf{U}^{\star\top}\|_F + \|\mathbf{X}\|_F)}\right\}$$

*then, we have*

$$\sup_{\mathbf{U}\in\mathcal{B}(l)}\|\operatorname{grad} f(\mathbf{U}) - \operatorname{grad} g(\mathbf{U})\|_F \le \frac{\epsilon}{2}, \quad \text{and} \quad \sup_{\mathbf{U}\in\mathcal{B}(l)}\|\operatorname{hess} f(\mathbf{U}) - \operatorname{hess} g(\mathbf{U})\|_2 \le \frac{\eta}{2}.$$

The proof of Lemma 3.2 is given in Appendix E (see supplementary material). As is shown in existing literature [25, 26, 27], a Gaussian linear operator $\mathcal{B} : \mathbb{R}^{N\times N} \to \mathbb{R}^M$ satisfies the RIP condition (3.4) with high probability if $M \ge C(r + k)N/\delta_{r+k}^2$ for some numerical constant $C$. Therefore, we can conclude that the three statements in Theorem 2.1 hold for the empirical risk (3.2) and population risk (3.3) as long as $M$ is large enough. Some similar bounds for the sample complexity $M$ under different settings can also be found in papers [8, 4]. Note that the particular choice of $l$ can guarantee that $\|\operatorname{grad} f(\mathbf{U})\|_F$ is large outside of $\mathcal{B}(l)$, which is also proved in Appendix E. Together with Theorem 2.1, we prove a globally benign landscape for the empirical risk.

### 3.2 Phase Retrieval

We continue to elaborate on Example 1.2. The following lemma provides the global geometry of the population risk in (1.2), which also determines the values of $\epsilon$ and $\eta$ in Assumption 2.1.

**Lemma 3.3.** *Define the following four regions:*

$$\mathcal{R}_1 \triangleq \left\{\boldsymbol{x} \in \mathbb{R}^N : \|\boldsymbol{x}\|_2 \le \frac{1}{2}\|\boldsymbol{x}^\star\|_2\right\}, \quad \mathcal{R}_2 \triangleq \left\{\boldsymbol{x} \in \mathbb{R}^N : \min_{\gamma\in\{1,-1\}}\|\boldsymbol{x} - \gamma\boldsymbol{x}^\star\|_2 \le \frac{1}{10}\|\boldsymbol{x}^\star\|_2\right\},$$

$$\mathcal{R}_3 \triangleq \left\{\boldsymbol{x} \in \mathbb{R}^N : \min_{\gamma\in\{1,-1\}}\left\|\boldsymbol{x} - \gamma\frac{1}{\sqrt{3}}\|\boldsymbol{x}^\star\|_2\boldsymbol{w}\right\|_2 \le \frac{1}{5}\|\boldsymbol{x}^\star\|_2, \ \boldsymbol{w}^\top\boldsymbol{x}^\star = 0, \ \|\boldsymbol{w}\|_2 = 1\right\},$$

$$\mathcal{R}_4 \triangleq \left\{\boldsymbol{x} \in \mathbb{R}^N : \|\boldsymbol{x}\|_2 > \frac{1}{2}\|\boldsymbol{x}^\star\|_2, \ \min_{\gamma\in\{1,-1\}}\|\boldsymbol{x} - \gamma\boldsymbol{x}^\star\|_2 > \frac{1}{10}\|\boldsymbol{x}^\star\|_2,\right.$$

$$\left.\min_{\gamma\in\{1,-1\}}\left\|\boldsymbol{x} - \gamma\frac{1}{\sqrt{3}}\|\boldsymbol{x}^\star\|_2\boldsymbol{w}\right\|_2 > \frac{1}{5}\|\boldsymbol{x}^\star\|_2, \ \boldsymbol{w}^\top\boldsymbol{x}^\star = 0, \ \|\boldsymbol{w}\|_2 = 1\right\}$$

*Then, the following properties hold:*

*(1)* $\boldsymbol{x} = \boldsymbol{0}$ *is a strict saddle point with* $\nabla^2 g(\boldsymbol{0}) = -4\boldsymbol{x}^\star\boldsymbol{x}^{\star\top} - 2\|\boldsymbol{x}^\star\|_2^2\mathbf{I}_N$ *and* $\lambda_{\min}(\nabla^2 g(\boldsymbol{0})) = -6\|\boldsymbol{x}^\star\|_2^2$. *Moreover, for any* $\boldsymbol{x} \in \mathcal{R}_1$, *the neighborhood of strict saddle point* $\boldsymbol{0}$, *we have*

$$\lambda_{\min}(\nabla^2 g(\boldsymbol{x})) \leq -\frac{3}{2}\|\boldsymbol{x}^\star\|_2^2.$$

*(2)* $\boldsymbol{x} = \pm\boldsymbol{x}^\star$ *are global minima with* $\nabla^2 g(\pm\boldsymbol{x}^\star) = 8\boldsymbol{x}^\star\boldsymbol{x}^{\star\top} + 4\|\boldsymbol{x}^\star\|_2^2\mathbf{I}_N$ *and* $\lambda_{\min}(\nabla^2 g(\pm\boldsymbol{x}^\star)) = 4\|\boldsymbol{x}^\star\|_2^2$. *Moreover, for any* $\boldsymbol{x} \in \mathcal{R}_2$, *the neighborhood of global minima* $\pm\boldsymbol{x}^\star$, *we have*

$$\lambda_{\min}(\nabla^2 g(\boldsymbol{x})) \geq 0.22\|\boldsymbol{x}^\star\|_2^2.$$

*(3)* $\boldsymbol{x} = \pm\frac{1}{\sqrt{3}}\|\boldsymbol{x}^\star\|_2\boldsymbol{w}$, *with* $\boldsymbol{w}^\top\boldsymbol{x}^\star = 0$ *and* $\|\boldsymbol{w}\|_2 = 1$, *are strict saddle points with* $\nabla^2 g(\pm\frac{1}{\sqrt{3}}\|\boldsymbol{x}^\star\|_2\boldsymbol{w}) = 4\|\boldsymbol{x}^\star\|_2^2\boldsymbol{w}\boldsymbol{w}^\top - 4\boldsymbol{x}^\star\boldsymbol{x}^{\star\top}$ *and* $\lambda_{\min}(\nabla^2 g(\pm\frac{1}{\sqrt{3}}\|\boldsymbol{x}^\star\|_2\boldsymbol{w})) = -4\|\boldsymbol{x}^\star\|_2^2$. *Moreover, for any* $\boldsymbol{x} \in \mathcal{R}_3$, *the neighborhood of strict saddle points* $\pm\frac{1}{\sqrt{3}}\|\boldsymbol{x}^\star\|_2\boldsymbol{w}$, *we have*

$$\lambda_{\min}(\nabla^2 g(\boldsymbol{x})) \leq -0.78\|\boldsymbol{x}^\star\|_2^2.$$

*(4) For any* $\boldsymbol{x} \in \mathcal{R}_4$, *the complement region of* $\mathcal{R}_1$, $\mathcal{R}_2$, *and* $\mathcal{R}_3$, *we have* $\|\nabla g(\boldsymbol{x})\|_2 > 0.3963\|\boldsymbol{x}^\star\|_2^3$.

The proof of Lemma 3.3 is inspired by the proof of [8, Theorem 3] and is given in Appendix F (see supplementary material). Letting $\epsilon \leq 0.3963\|\boldsymbol{x}^\star\|_2^3$ and $\eta = 0.22\|\boldsymbol{x}^\star\|_2^2$, the population risk (1.2) then satisfies Assumption 2.1. As in Lemma 3.1, we also note that each critical point of the population risk in (1.2) is either a global minimum or a strict saddle. This inspires us to carry this favorable geometry over to the corresponding empirical risk.

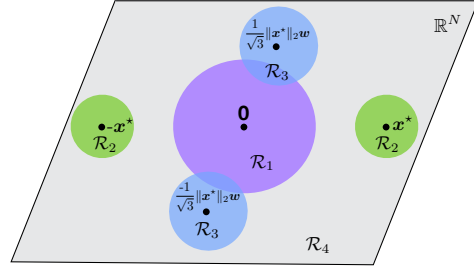

Figure 3: Partition of regions in Lemma 3.3.

The partition of regions used in Lemma 3.3 is illustrated in Figure 3. We use the purple, green, and blue balls to denote the three regions $\mathcal{R}_1$, $\mathcal{R}_2$, and $\mathcal{R}_3$, respectively. $\mathcal{R}_4$ is then represented with the light gray region. Therefore, the union of the four regions covers the entire $\mathbb{R}^N$ space.

Define a norm ball as $\mathcal{B}(l) \triangleq \{\boldsymbol{x} \in \mathbb{R}^N : \|\boldsymbol{x}\|_2 \leq l\}$ with radius $l = 1.1\|\boldsymbol{x}^\star\|_2$. This particular choice of $l$ guarantees that $\|\operatorname{grad} f(\boldsymbol{x})\|_2$ is large outside of $\mathcal{B}(l)$, which is proved in Appendix G. Together with Theorem 2.1, we prove a globally benign landscape for the empirical risk. We also define $h(N, M) \triangleq \widetilde{\mathcal{O}}\left(\frac{N^2}{M} + \sqrt{\frac{N}{M}}\right)$ with $\widetilde{\mathcal{O}}$ denoting an asymptotic notation that hides polylog factors. The following lemma verifies Assumptions 2.2 and 2.3 for this phase retrieval problem.

**Lemma 3.4.** *Suppose that* $\boldsymbol{a}_m \in \mathbb{R}^N$ *is a Gaussian random vector with entries following* $\mathcal{N}(0, 1)$. *If* $h(N, M) \leq 0.0118$, *we then have*

$$\sup_{\boldsymbol{x}\in\mathcal{B}(l)} \|\nabla f(\boldsymbol{x}) - \nabla g(\boldsymbol{x})\|_2 \leq \frac{\epsilon}{2}, \quad \text{and} \quad \sup_{\boldsymbol{x}\in\mathcal{B}(l)} \|\nabla^2 f(\boldsymbol{x}) - \nabla^2 g(\boldsymbol{x})\|_2 \leq \frac{\eta}{2}$$

*hold with probability at least* $1 - e^{-CN\log(M)}$.

The proof of Lemma 3.4 is given in Appendix G (see supplementary material). The assumption $h(N, M) \leq 0.0118$ implies that we need a sample complexity that scales like $N^2$, which is not optimal since $\boldsymbol{x}$ has only $N$ degrees of freedom. This is a technical artifact that can be traced back to Assumptions 2.2 and 2.3–which require two-sided closeness between the gradients and Hessians–and the heavy-tail property of the fourth powers of Gaussian random process [12]. To arrive at the conclusions of Theorem 2.1, however, these two assumptions are sufficient but not necessary (while Assumption 2.1 is more critical), leaving room for tightening the sampling complexity bound. We leave this to future work.

# 4 Numerical Simulations

We first conduct numerical experiments on the two examples introduced in Section 1, i.e., the rank-1 matrix sensing and phase retrieval problems. In both problems, we fix $N = 2$ and set $\boldsymbol{x}^\star = [1 \ -1]^\top$.

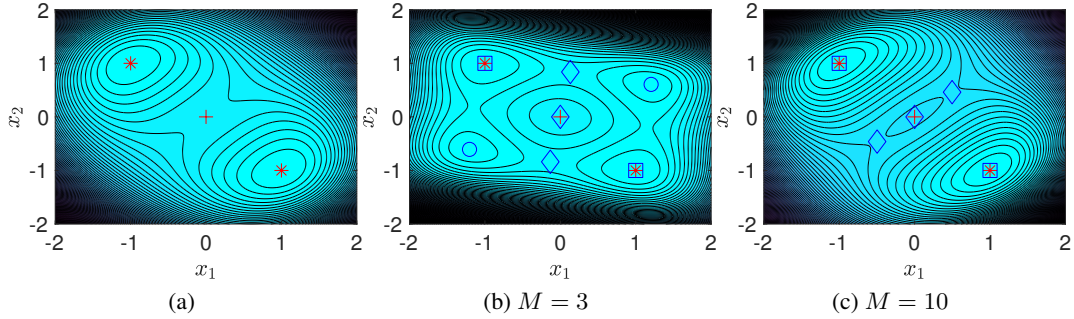

Figure 4: Rank-1 matrix sensing: (a) Population risk. (b, c) A realization of empirical risk. In both this and Figure 5, we use the red star and cross to denote the global minima and saddle points of the population risk, and use blue square, circle, and diamond to denote the global minima, spurious local minima, and saddle points of the empirical risk, respectively.

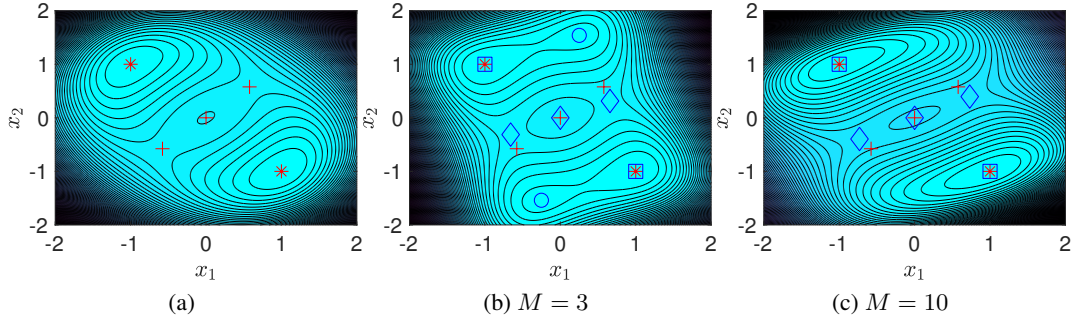

Figure 5: Phase retrieval: (a) Population risk. (b, c) A realization of empirical risk.

Then, we generate the population risk and empirical risk based on the formulation introduced in these two examples. The contour plots of the population risk and a realization of empirical risk with $M = 3$ and $M = 10$ are given in Figure 4 for rank-1 matrix sensing and Figure 5 for phase retrieval. We see that when we have fewer samples (e.g., $M = 3$), there could exist some spurious local minima as is shown in plots (b). However, as we increase the number of samples (e.g., $M = 10$), we see a direct correspondence between the local

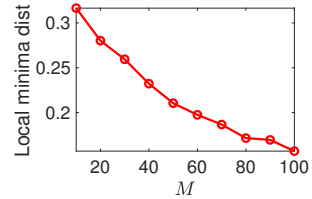

Figure 6: Rank-2 matrix sensing.

minima of empirical risk and population risk in both examples with a much higher probability. We also notice that extra saddle points can emerge as shown in Figure 4 (c), which shows that statement (b) in Theorem 2.1 cannot be improved to a one-to-one correspondence between saddle points in degenerate scenarios. We still observe this phenomenon even when $M = 1000$, which is not shown here. Note that for the rank-1 case, Theorem 2.1 can be applied directly without restricting to full-rank representations. Next, we conduct another experiment on general-rank matrix sensing with $k = 2$, $r = 3$, $N = 8$, and a variety of $M$. We set $\mathbf{U}^\star$ as the first $r$ columns of an $N \times N$ identity matrix and create $\mathbf{X} = \mathbf{U}^\star \mathbf{U}^{\star\top}$. The population and empirical risks are then generated according to the model introduced in Section 3.1. As shown in Figure 6, the distance (averaged over 100 trials) between the local minima of the population and empirical risk decreases as we increase $M$.

## 5   Conclusions

In this work, we study the problem of establishing a correspondence between the critical points of the empirical risk and its population counterpart without the strongly Morse assumption required in some existing literature. With this correspondence, we are able to analyze the landscape of an empirical risk from the landscape of its population risk. Our theory builds on a weaker condition than the strongly Morse assumption. This enables us to work on the very popular matrix sensing and phase retrieval problems, whose Hessian does have zero eigenvalues at some critical points, i.e., they are degenerate and do not satisfy the strongly Morse assumption. As mentioned, there is still room to improve the sample complexity of the phase retrieval problem that we will pursue in future work.

**Acknowledgments**

SL would like to thank Qiuwei Li at Colorado School of Mines for many helpful discussions on the analysis of matrix sensing and phase retrieval. The authors would also like to thank the anonymous reviewers for their constructive comments and suggestions which greatly improved the quality of this paper. This work was supported by NSF grant CCF-1704204, and the DARPA Lagrange Program under ONR/SPAWAR contract N660011824020.

## Footnotes

[1]A twice differentiable function $f(\boldsymbol{x})$ is *Morse* if all of its critical points are non-degenerate, i.e., its Hessian has no zero eigenvalues at all critical points. Mathematically, $\nabla f(\boldsymbol{x}) = \boldsymbol{0}$ implies all $\lambda_i(\nabla^2 f(\boldsymbol{x})) \neq 0$ with $\lambda_i(\cdot)$ being the $i$-th eigenvalue of the Hessian. A twice differentiable function $f(\boldsymbol{x})$ is $(\epsilon, \eta)$-*strongly Morse* if $\|\nabla f(\boldsymbol{x})\|_2 \leq \epsilon$ implies $\min_i |\lambda_i(\nabla^2 f(\boldsymbol{x}))| \geq \eta$. One can refer to [1] for more information.

[2]The subset $\mathcal{B}(l)$ can vary in different applications. For example, we define $\mathcal{B}(l) \triangleq \{\mathbf{U} \in \mathbb{R}_*^{N \times k} : \|\mathbf{U}\mathbf{U}^\top\|_F \leq l\}$ in matrix sensing and $\mathcal{B}(l) \triangleq \{\boldsymbol{x} \in \mathbb{R}^N : \|\boldsymbol{x}\|_2 \leq l\}$ in phase retrieval.

[3]For problems with matrix variables, such as matrix sensing introduced in Section 3, $\boldsymbol{x}$ is the vectorized representation of the matrix.

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
