[Supplementary Material · NIPS19_PR_Supp_final3.pdf]

# Supplementary Material for Paper "The Landscape of Non-convex Empirical Risk with Degenerate Population Risk"

**Shuang Li, Gongguo Tang, and Michael B. Wakin**
Department of Electrical Engineering
Colorado School of Mines
Golden, CO 80401
{shuangli,gtang,mwakin}@mines.edu

## A  Proof of Theorem 2.1

To prove Theorem 2.1, we need the following two lemmas, which are extensions of [1, Lemmas 5, 7].

**Lemma A.1.** *Let $\mathcal{M}$ be a general Riemannian manifold and $\mathcal{E} \subseteq \mathcal{M}$ be a connected and compact set with a $\mathcal{C}^2$ boundary $\partial\mathcal{E}$. Denote $f, g : \mathcal{A}_o \to \mathbb{R}$ as two $\mathcal{C}^2$ functions defined on an open set $\mathcal{A}_o$ with $\mathcal{E} \subseteq \mathcal{A}_o \subseteq \mathcal{M}$. With the following assumptions:*

- *For all $\boldsymbol{x} \in \partial\mathcal{E}$ and $t \in [0,1]$,*

$$t\,\mathrm{grad}\, f(\boldsymbol{x}) + (1-t)\mathrm{grad}\, g(\boldsymbol{x}) \neq \boldsymbol{0}. \tag{A.1}$$

- *The Hessians of $f$ and $g$ are close, i.e.,*

$$\|\mathrm{hess}\, f(\boldsymbol{x}) - \mathrm{hess}\, g(\boldsymbol{x})\|_2 \leq \frac{\eta}{2}. \tag{A.2}$$

- *For all $\boldsymbol{x} \in \mathcal{E}$, the minimal eigenvalue of hess $g(\boldsymbol{x})$ satisfies*

$$|\lambda_{\min}(\mathrm{hess}\, g(\boldsymbol{x}))| \geq \eta. \tag{A.3}$$

*Then, we have the following statements hold:*

*(a) Both $g$ and $f$ have at most a finite number of local minima in $\mathcal{E}$. Furthermore, if $g$ has $K$ ($K = 0, 1, 2, \cdots$) local minima in $\mathcal{E}$, then $f$ also has $K$ local minima in $\mathcal{E}$.*

*(b) If $g$ has a strict saddle in $\mathcal{E}$, then if $f$ has any critical points in $\mathcal{E}$, they must be strict saddle points.*

The proof of Lemma A.1 is given in Appendix B.

The following lemma is a parallel result of [1, Lemma 7] for the case when

$$\lambda_{\min}(\mathrm{hess}\, g(\boldsymbol{x})) \geq \eta, \ \lambda_{\min}(\mathrm{hess}\, f(\boldsymbol{x})) \geq \frac{\eta}{2},$$

and can be proved similarly.

**Lemma A.2.** *Denote $\mathcal{B}(l)$ as a compact and connected subset in a general manifold $\mathcal{M}$ with $l$ being its parameters.[1] Let $g : \mathcal{B}(l) \to \mathbb{R}$ be a $\mathcal{C}^2$ function satisfying $\lambda_{\min}(\mathrm{hess}\, g(\boldsymbol{x})) \geq \eta$ in $\overline{\mathcal{D}}$ with*

$\overline{\mathcal{D}} \triangleq \{\boldsymbol{x} \in \mathcal{B}(l) : \|\text{grad } g(\boldsymbol{x})\|_2 \leq \epsilon\}$. *Denote* $\boldsymbol{x}_1,\ \boldsymbol{x}_2,\ \cdots,\ \boldsymbol{x}_K$ *as the local minima of function g. Then, there exist disjoint compact sets* $\{\mathcal{D}_i\}_{i \in \mathbb{N}}$ *such that*

$$\overline{\mathcal{D}} = \cup_{i=1}^{\infty} \mathcal{D}_i$$

*with each maximal connected component* $\mathcal{D}_i$ *containing at most one local minimum. Namely,* $\boldsymbol{x}_i \in \mathcal{D}_i$ *for* $1 \leq i \leq K$, *and* $\mathcal{D}_i$ *with* $i \geq K + 1$ *contains no local minima.*

Now, we are ready to prove Theorem 2.1. Denote $\boldsymbol{x}_1, \cdots, \boldsymbol{x}_K$ as the $K$ local minima of $g(\boldsymbol{x})$. Define $\overline{\mathcal{D}} \triangleq \{\boldsymbol{x} \in \mathcal{B}(l) : \|\text{grad } g(\boldsymbol{x})\|_2 \leq \epsilon\}$. By applying Lemma A.2, we can partition $\overline{\mathcal{D}}$ as $\overline{\mathcal{D}} = \cup_{i=1}^{\infty} \mathcal{D}_i$, where each $\mathcal{D}_i$ is a disjoint connected compact component containing at most one local minimum. Explicitly, $\boldsymbol{x}_i \in \mathcal{D}_i$ for $1 \leq i \leq K$, and $\mathcal{D}_i$ with $i \geq K + 1$ contains no local minima. We also have $\|\text{grad } g(\boldsymbol{x})\|_2 = \epsilon$ for $\boldsymbol{x} \in \partial \mathcal{D}_i$ by the continuity of grad $g(\boldsymbol{x})$.

Hereafter, we assume the two Assumptions 2.2 and 2.3 hold. It follows from (2.2) that

$$\sup_{\boldsymbol{x} \in \partial \mathcal{D}_i} \|\text{grad } f(\boldsymbol{x}) - \text{grad } g(\boldsymbol{x})\|_2 \leq \frac{\epsilon}{2}.$$

Then, for $\forall\, t \in [0, 1]$, we have

$$\sup_{\boldsymbol{x} \in \partial \mathcal{D}_i} t\|\text{grad } f(\boldsymbol{x}) - \text{grad } g(\boldsymbol{x})\|_2 \leq \frac{\epsilon}{2},$$

which is equivalent to

$$\epsilon - \sup_{\boldsymbol{x} \in \partial \mathcal{D}_i} t\|\text{grad } f(\boldsymbol{x}) - \text{grad } g(\boldsymbol{x})\|_2 \geq \frac{\epsilon}{2}, \quad \forall\, t \in [0, 1].$$

Recall that $\|\text{grad } g(\boldsymbol{x})\|_2 = \epsilon$ for $\boldsymbol{x} \in \partial \mathcal{D}_i$. Then, we have

$$\inf_{\boldsymbol{x} \in \partial \mathcal{D}_i} \|\text{grad } g(\boldsymbol{x})\|_2 - \sup_{\boldsymbol{x} \in \partial \mathcal{D}_i} t\|\text{grad } f(\boldsymbol{x}) - \text{grad } g(\boldsymbol{x})\|_2 \geq \frac{\epsilon}{2}, \quad \forall\, t \in [0, 1],$$

which further gives us

$$\inf_{\boldsymbol{x} \in \partial \mathcal{D}_i} \{\|\text{grad } g(\boldsymbol{x})\|_2 - t\|\text{grad } f(\boldsymbol{x}) - \text{grad } g(\boldsymbol{x})\|_2\} \geq \frac{\epsilon}{2}, \quad \forall\, t \in [0, 1].$$

Consequently, we obtain

$$\inf_{\boldsymbol{x} \in \partial \mathcal{D}_i} \|(1 - t)\text{grad } g(\boldsymbol{x}) + t\text{grad } f(\boldsymbol{x})\|_2 \geq \frac{\epsilon}{2}, \quad \forall\, t \in [0, 1].$$

Let $\mathcal{D}$ in the statement of Theorem 2.1 be one of the $\mathcal{D}_i$s. Then $\mathcal{D}$ contains at most one local minimum. The rest of Theorem 2.1 follows from Lemma A.1.

## B  Proof of Lemma A.1

Using the Nash embedding theorem [2], we first embed the Riemannian manifold $\mathcal{M}$ isometrically into a Euclidean space $\mathbb{R}^{\bar{N}}$ for sufficiently large $\bar{N}$. This allows us to view $\mathcal{M}$ as a Riemannian submanifold of $\mathbb{R}^{\bar{N}}$ and identify the tangent spaces of $\mathcal{M}$ as subspaces of $\mathbb{R}^{\bar{N}}$. We also identify the norm $\|\cdot\|_2$ induced by the Riemannian metric with the Euclidean norm in $\mathbb{R}^{\bar{N}}$. Recall that $\mathcal{E}$ is a connected set. Then, assumption (A.3) implies that any point $\boldsymbol{x} \in \mathcal{E}$ satisfy either $\lambda_{\min}(\text{hess } g(\boldsymbol{x})) \geq \eta$ or $\lambda_{\min}(\text{hess } g(\boldsymbol{x})) \leq -\eta$. There cannot exist two points $\boldsymbol{x}_1, \boldsymbol{x}_2 \in \mathcal{E}$ such that $\lambda_{\min}(\text{hess } g(\boldsymbol{x}_1)) \geq \eta$ and $\lambda_{\min}(\text{hess } g(\boldsymbol{x}_2)) \leq -\eta$. Otherwise, since the continuous image of any connected set must also be a connected set, there must exist another point $\boldsymbol{x}_3 \in \mathcal{E}$ such that $-\eta < \lambda_{\min}(\text{hess } g(\boldsymbol{x}_3)) < \eta$, which contradicts assumption (A.3).

Note that

$$|\lambda_{\min}(\text{hess } f(\boldsymbol{x})) - \lambda_{\min}(\text{hess } g(\boldsymbol{x}))| \leq \|\text{hess } f(\boldsymbol{x}) - \text{hess } g(\boldsymbol{x})\|_2 \leq \frac{\eta}{2},$$

where the first inequality follows from [3, Theorem 5] and the last inequality follows from assumption (A.2). Together with the assumption (A.3), we obtain

$$\begin{cases} \lambda_{\min}(\text{hess } f(\boldsymbol{x})) \geq \frac{\eta}{2}, & \text{if } \lambda_{\min}(\text{hess } g(\boldsymbol{x})) \geq \eta, \\ \lambda_{\min}(\text{hess } f(\boldsymbol{x})) \leq -\frac{\eta}{2}, & \text{if } \lambda_{\min}(\text{hess } g(\boldsymbol{x})) \leq -\eta. \end{cases} \tag{B.1}$$

1) When $\lambda_{\min}(\text{hess } g(\boldsymbol{x})) \geq \eta$ for all $\boldsymbol{x} \in \mathcal{E}$, we have $\lambda_{\min}(\text{hess } f(\boldsymbol{x})) \geq \frac{\eta}{2}$ for all $\boldsymbol{x} \in \mathcal{E}$. This implies that the critical points of $g(\boldsymbol{x})$ and $f(\boldsymbol{x})$ in $\mathcal{E}$ are all local minima and are all isolated. Since $\mathcal{E}$ is a compact set, there can only exist a finite number of critical points of $g(\boldsymbol{x})$ and $f(\boldsymbol{x})$ in $\mathcal{E}$, which are denoted as $\boldsymbol{x}_1, \boldsymbol{x}_2, \cdots, \boldsymbol{x}_K$ and $\widehat{\boldsymbol{x}}_1, \widehat{\boldsymbol{x}}_2, \cdots, \widehat{\boldsymbol{x}}_{\widehat{K}}$, respectively.

For $\epsilon > 0$ small enough, define a set

$$\mathcal{E}_{-\epsilon} \triangleq \{\boldsymbol{x} \in \mathcal{E} : d(\boldsymbol{x}, \mathcal{E}^c) \geq \epsilon\},$$

where $d(\boldsymbol{x}, \mathcal{S}) \triangleq \inf\{\|\boldsymbol{x} - \boldsymbol{y}\|_2 : \boldsymbol{y} \in \mathcal{S}\}$ is the distance between $\boldsymbol{x}$ and a set $\mathcal{S}$. Define $w : \mathcal{A}_o \to [0, 1]$ as a $\mathcal{C}^1$ bump function with

$$w(\boldsymbol{x}) = \begin{cases} 0, & \boldsymbol{x} \in \mathcal{A}_o \backslash \mathcal{E}, \\ 1, & \boldsymbol{x} \in \mathcal{E}_{-\epsilon}. \end{cases}$$

Define two $\mathcal{C}^1$ vector fields as

$$\boldsymbol{\xi}_0(\boldsymbol{x}) = \text{grad } g(\boldsymbol{x}),$$
$$\boldsymbol{\xi}_1(\boldsymbol{x}) = (1 - w(\boldsymbol{x}))\text{grad } g(\boldsymbol{x}) + w(\boldsymbol{x})\text{grad } f(\boldsymbol{x}).$$

Note that $\boldsymbol{\xi}_0|_{\partial \mathcal{E}} = \boldsymbol{\xi}_1|_{\partial \mathcal{E}}$ since $w(\boldsymbol{x}) = 0$ when $\boldsymbol{x} \in \partial \mathcal{E}$. With assumption (A.1), we have

$$\inf_{\boldsymbol{x} \in \partial \mathcal{E}} \inf_{t \in [0,1]} \|(1 - t)\text{grad } g(\boldsymbol{x}) + t\text{grad } f(\boldsymbol{x})\|_2 > 0$$

by a continuity argument. Then, we can choose $\epsilon > 0$ small enough such that

$$\boldsymbol{\xi}_1(\boldsymbol{x}) \neq 0, \quad \text{hess } f(\boldsymbol{x}) \neq 0$$

holds for all $\boldsymbol{x} \in \mathcal{E} \backslash \mathcal{E}_{-\epsilon}$. This implies that the critical points of $\boldsymbol{\xi}_1{}^2$ are all in $\mathcal{E}_{-\epsilon}$ and coincide with the critical points of $f$ since $\boldsymbol{\xi}_1(\boldsymbol{x}) = \text{grad } f(\boldsymbol{x})$ in $\mathcal{E}_{-\epsilon}$. Therefore, $\widehat{\boldsymbol{x}}_1, \widehat{\boldsymbol{x}}_2, \cdots, \widehat{\boldsymbol{x}}_{\widehat{K}}$ are also the critical points of $\boldsymbol{\xi}_1$ in $\mathcal{E}_{-\epsilon}$.

For a non-degenerate critical point $\boldsymbol{x}_0$ of a smooth vector field $\boldsymbol{\xi} : \mathcal{E} \to \mathbb{R}^{\bar{N}}$, we define the index of $\boldsymbol{x}_0$ as the sign of the Jacobian determinant [1, 4], namely

$$\text{ind}_{\boldsymbol{x}_0}(\boldsymbol{\xi}) = \text{sign} \ \det \left(\text{D}\boldsymbol{\xi}_{\boldsymbol{x}_0}\right), \tag{B.2}$$

where $\text{D}\boldsymbol{\xi}_{\boldsymbol{x}_0} : T_{\boldsymbol{x}_0}\mathcal{M} \to \mathbb{R}^{\bar{N}}$ is the differential of the vector field. Note that the map $\text{D}\boldsymbol{\xi}_{\boldsymbol{x}_0}$ can be considered as a linear transformation from $T_{\boldsymbol{x}_0}\mathcal{M}$ to itself and hence has a well-defined determinant. When $\boldsymbol{\xi}$ is the Riemannian gradient, the differential $\text{D}\boldsymbol{\xi}_{\boldsymbol{x}_0}$ reduces to the Riemannian Hessian [5, Definition 5.5.1 and equation (5.15)].

Since $\lambda_{\min}(\text{hess } g(\boldsymbol{x})) \geq \eta$ and $\lambda_{\min}(\text{hess } f(\boldsymbol{x})) \geq \frac{\eta}{2}$, both hess $g(\boldsymbol{x})$ and hess $f(\boldsymbol{x})$ are non-degenerate matrices whose determinants are positive. Recall that $\boldsymbol{\xi}_1(\boldsymbol{x}) = \text{grad } f(\boldsymbol{x})$ when $\boldsymbol{x} \in \mathcal{E}_{-\epsilon}$. Then, for $1 \leq i \leq \widehat{K}$, we have

$$\text{ind}_{\widehat{\boldsymbol{x}}_i}(\boldsymbol{\xi}_1) = \text{sign} \ \det (\text{D}(\boldsymbol{\xi}_1)_{\widehat{\boldsymbol{x}}_i}) = \text{sign} \ \det (\text{hess } f(\widehat{\boldsymbol{x}}_i)) = 1.$$

Define $\widehat{\boldsymbol{\xi}}(\boldsymbol{x}) \triangleq \boldsymbol{\xi}(\boldsymbol{x})/\|\boldsymbol{\xi}(\boldsymbol{x})\|_2$ wherever $\boldsymbol{\xi}(\boldsymbol{x}) \neq \boldsymbol{0}$ as the Gauss map. Denote $\boldsymbol{x}_1, \boldsymbol{x}_2, \cdots, \boldsymbol{x}_K$ as the critical points of function $g$ in $\mathcal{E}$. It follows from [1, Lemma 6], [6, Theorem 1.1.2], and [4, Theorem 14.4.4] that the sum of indices of the critical points inside $\mathcal{E}$ is equal to the degree of the Gauss map restricted to the boundary of $\mathcal{E}$, hence, we have

$$\widehat{K} = \sum_{i=1}^{\widehat{K}} \text{ind}_{\widehat{\boldsymbol{x}}_i}(\boldsymbol{\xi}_1) = \deg \left(\widehat{\boldsymbol{\xi}}_1|_{\partial \mathcal{E}}\right) \overset{①}{=} \deg \left(\widehat{\boldsymbol{\xi}}_0|_{\partial \mathcal{E}}\right)$$

$$= \sum_{i=1}^{K} \text{ind}_{\boldsymbol{x}_i}(\boldsymbol{\xi}_0) \overset{②}{=} \sum_{i=1}^{K} \text{sign} \ \det (\text{D}(\boldsymbol{\xi}_0)_{\boldsymbol{x}_i})$$

$$= \sum_{i=1}^{K} \text{sign} \ \det (\text{hess } g(\boldsymbol{x}_i)) = K,$$

---

²For a smooth vector field $\boldsymbol{\xi} : \mathcal{E} \to T\mathcal{M}$, defined on $\mathcal{E} \subseteq \mathcal{M}$, a critical point is defined as a point $\boldsymbol{x}_0 \in \mathcal{E}$ satisfying $\boldsymbol{\xi}(\boldsymbol{x}_0) = \boldsymbol{0}$. Here $T\mathcal{M}$ is the tangent bundle of $\mathcal{M}$.

where $\deg\left(\widehat{\boldsymbol{\xi}}|_{\partial\mathcal{E}}\right)$ denotes the degree of the Gauss map restricted to the boundary of $\mathcal{E}$. Here, ① follows from $\boldsymbol{\xi}_0|_{\partial\mathcal{E}} = \boldsymbol{\xi}_1|_{\partial\mathcal{E}}$ and ② follows from (B.2). Then, we can conclude that the number of critical points of $f$ and $g$ are both equal to $K = \widehat{K}$. Since the minimal eigenvalues of $g$ and $f$ are both positive, the critical points are also local minima. Thus, we finish the proof for first part of Lemma A.1.

2) When $\lambda_{\min}(\mathrm{hess}\ g(\boldsymbol{x})) \leq -\eta$, we have $\lambda_{\min}(\mathrm{hess}\ f(\boldsymbol{x})) \leq -\frac{\eta}{2}$. This immediately implies the second part of Lemma A.1.

## C    Proof of Corollary 2.1

Let $\{\widehat{\boldsymbol{x}}_k\}_{k=1}^K$ and $\{\boldsymbol{x}_k\}_{k=1}^K$ denote the local minima of the empirical risk $f$ and its population risk $g$. Recall that $\overline{\mathcal{D}} = \{\boldsymbol{x} \in \mathcal{B}(l) : \|\mathrm{grad}\ g(\boldsymbol{x})\|_2 \leq \epsilon\}$. Using Lemma A.2, we partition $\overline{\mathcal{D}}$ as $\overline{\mathcal{D}} = \cup_{k=1}^\infty \mathcal{D}_k$ with $\boldsymbol{x}_k, \widehat{\boldsymbol{x}}_k \in \mathcal{D}_k$ for $1 \leq k \leq K$, and $\mathcal{D}_k$ for $k \geq K + 1$ contains no local minima.

Fix $k \in \{1, 2, \ldots, K\}$. Let $\mathcal{T}_{\boldsymbol{x}_k}\mathcal{M}$ be the tangent space of the Riemannian manifold $\mathcal{M}$ at $\boldsymbol{x}_k$ and $\boldsymbol{0}_{\boldsymbol{x}_k}$ be the zero vector of $\mathcal{T}_{\boldsymbol{x}_k}\mathcal{M}$. Let $\mathrm{Exp}_{\boldsymbol{x}_k} : \mathcal{T}_{\boldsymbol{x}_k}\mathcal{M} \to \mathcal{M}$ denote the *exponential map* at $\boldsymbol{x}_k$. Suppose $\widehat{\mathcal{N}}_{\boldsymbol{x}_k}$ is an open ball in $\mathcal{T}_{\boldsymbol{x}_k}\mathcal{M}$ around $\boldsymbol{0}_{\boldsymbol{x}_k}$ with radius $\rho$, the injectivity radius of $\mathcal{M}$. Then $\mathrm{Exp}_{\boldsymbol{x}_k}$ is a diffeomorphism in $\widehat{\mathcal{N}}_{\boldsymbol{x}_k}$ [5, pp.148-149]. Define $\mathcal{N}_{\boldsymbol{x}_k} \triangleq \mathrm{Exp}_{\boldsymbol{x}_k}(\widehat{\mathcal{N}}_{\boldsymbol{x}_k})$ as the image of $\widehat{\mathcal{N}}_{\boldsymbol{x}_k}$ under the exponential map $\mathrm{Exp}_{\boldsymbol{x}_k}$. Then the Riemannian distance

$$\mathrm{dist}(\boldsymbol{z}_1, \boldsymbol{z}_2) = \|\mathrm{Exp}_{\boldsymbol{x}_k}^{-1}(\boldsymbol{z}_1) - \mathrm{Exp}_{\boldsymbol{x}_k}^{-1}(\boldsymbol{z}_2)\|_2, \forall \boldsymbol{z}_1, \boldsymbol{z}_2 \in \mathcal{N}_{\boldsymbol{x}_k}$$

is equivalent to the distance in the tangent space (induced by the Riemannian metric) [5, Section 4.5.1]. The corollary's assumptions ensure in particular that $\widehat{\boldsymbol{x}}_k \in \mathcal{D}_k \subseteq \mathcal{N}_{\boldsymbol{x}_k}$. We next bound the radius of the set $\mathcal{D}_k$.

Consider the *pullback* $\widehat{g} = g \circ \mathrm{Exp}_{\boldsymbol{x}_k} : \mathcal{T}_{\boldsymbol{x}_k}\mathcal{M} \to \mathbb{R}$ that "pulls back" the cost function $g$ from the manifold $\mathcal{M}$ to the vector space $\mathcal{T}_{\boldsymbol{x}_k}\mathcal{M}$. Since the exponential map is a retraction of at least second-order, the gradient and Hessian of the pullback[3] satisfy [7, Proposition 2.11, Corollary 2.13]

$$\nabla\widehat{g}(\boldsymbol{0}_{\boldsymbol{x}_k}) = \mathrm{grad}\ g(\boldsymbol{x}_k) = \boldsymbol{0}_{\boldsymbol{x}_k}, \quad \nabla^2\widehat{g}(\boldsymbol{0}_{\boldsymbol{x}_k}) = \mathrm{hess}\ g(\boldsymbol{x}_k).$$

This together with the Lipschitz Hessian condition imply that [8, Lemma 1]

$$\|\nabla\widehat{g}(\boldsymbol{v}) - \mathrm{hess}\ g(\boldsymbol{x}_k)[\boldsymbol{v}]\|_2 = \|\nabla\widehat{g}(\boldsymbol{v}) - \nabla\widehat{g}(\boldsymbol{0}_{\boldsymbol{x}_k}) - \nabla^2\widehat{g}(\boldsymbol{0}_{\boldsymbol{x}_k})[\boldsymbol{v}]\|_2 \leq \frac{L_H}{2}\|\boldsymbol{v}\|_2^2.$$

Since $\lambda_{\min}(\mathrm{hess}\ g(\boldsymbol{x}_k)) \geq \eta$, we conclude

$$\|\nabla\widehat{g}(\boldsymbol{v})\|_2 \geq \|\mathrm{hess}\ g(\boldsymbol{x}_k)[\boldsymbol{v}]\|_2 - \frac{L_H}{2}\|\boldsymbol{v}\|_2^2 \geq \eta\|\boldsymbol{v}\|_2 - \frac{L_H}{2}\|\boldsymbol{v}\|_2^2. \qquad (C.1)$$

Since the gradient of the pullback $\widehat{g}$ at $\boldsymbol{v}$ and the Riemannian gradient of $g$ at $\mathrm{Exp}_{\boldsymbol{x}_k}(\boldsymbol{v})$ satisfy [9, Lemma 5.2]

$$\nabla\widehat{g}(\boldsymbol{v}) = \left(\mathrm{DExp}_{\boldsymbol{x}_k}(\boldsymbol{v})\right)^* \left[\mathrm{grad}\ g\left(\mathrm{Exp}_{\boldsymbol{x}_k}(\boldsymbol{v})\right)\right],$$

where the differential $\mathrm{DExp}_{\boldsymbol{x}_k}(\boldsymbol{v})$ is a linear operator mapping vectors from the tangent space at $\boldsymbol{x}_k$ to the tangent space at $\mathrm{Exp}_{\boldsymbol{x}_k}(\boldsymbol{v})$, and the star indicates the adjoint, the corollary's assumptions imply

$$\|\nabla\widehat{g}(\boldsymbol{v})\|_2 \leq \|\mathrm{DExp}_{\boldsymbol{x}_k}(\boldsymbol{v})\|\|\mathrm{grad}\ g\left(\mathrm{Exp}_{\boldsymbol{x}_k}(\boldsymbol{v})\right)\|_2 \leq \sigma\|\mathrm{grad}\ g\left(\mathrm{Exp}_{\boldsymbol{x}_k}(\boldsymbol{v})\right)\|_2.$$

Combining this with (C.1) yields

$$\|\mathrm{grad}\ g\left(\mathrm{Exp}_{\boldsymbol{x}_k}(\boldsymbol{v})\right)\|_2 \geq \frac{\eta}{\sigma}\|\boldsymbol{v}\|_2 - \frac{L_H}{2\sigma}\|\boldsymbol{v}\|_2^2. \qquad (C.2)$$

Define $\widetilde{\mathcal{D}}_k \triangleq \{\boldsymbol{x} = \mathrm{Exp}_{\boldsymbol{x}_k}(\boldsymbol{v}) \in \mathcal{N}_{\boldsymbol{x}_k} : \frac{\eta}{\sigma}\|\boldsymbol{v}\|_2 - \frac{L_H}{2\sigma}\|\boldsymbol{v}\|_2^2 \leq \epsilon\}$. It follows from (C.2) that $\mathcal{D}_k \subseteq \widetilde{\mathcal{D}}_k$. Let $r_0 = \frac{\eta - \sqrt{\eta^2 - 2\sigma L_H \epsilon}}{L_H}$ and $r_1 = \frac{\eta + \sqrt{\eta^2 - 2\sigma L_H \epsilon}}{L_H}$. For $\epsilon \leq \eta^2/(2\sigma L_H)$, we have

$\widetilde{\mathcal{D}}_k = \mathcal{B}(r_0) \cup \mathcal{B}(r_1)^c$ with $\mathcal{B}(r_1)^c$ being the complement of $\mathcal{B}(r_1)$. Here $\mathcal{B}(r_0) = \{\boldsymbol{x} = \mathrm{Exp}_{\boldsymbol{x}_k}(\boldsymbol{v}) : \|\boldsymbol{v}\|_2 \leq r_0\} = \{\boldsymbol{x} \in \mathcal{M} : \mathrm{dist}(\boldsymbol{x}, \boldsymbol{x}_k) \leq r_0\}$. Note that since $\mathcal{D}_k$ is connected and $\boldsymbol{x}_k \in \mathcal{D}_k \cap \mathcal{B}(r_0)$, we then have $\mathcal{D}_k \subseteq \mathcal{B}(r_0)$, which together with $\widehat{\boldsymbol{x}}_k \in \mathcal{D}_k$ further indicates that

$$\mathrm{dist}(\widehat{\boldsymbol{x}}_k, \boldsymbol{x}_k) \leq r_0 \leq 2\sigma\epsilon/\eta,$$

where the last inequality follows from $\epsilon \leq \eta^2/(2\sigma L_H)$ and the elementary inequality $\sqrt{1-x} \geq 1-x$ for $x \in [0,1]$. This completes the proof since $k \in \{1, 2, \dots, K\}$ is arbitrary.

# D  Proof of Lemma 3.1

We present the Riemannian gradient and Hessian of population risk on the quotient manifold $\mathcal{M}$ as follows

$$\mathrm{grad}\, g(\mathbf{U}) = \mathcal{P}_{\mathbf{U}}(\nabla g(\mathbf{U})) = (\mathbf{U}\mathbf{U}^\top - \mathbf{X})\mathbf{U}$$

$$\mathrm{hess}\, g(\mathbf{U})[\mathbf{D}, \mathbf{D}] = \langle \mathcal{P}_{\mathbf{U}}(\nabla^2 g(\mathbf{U})[\mathbf{D}]), \mathbf{D}\rangle = \nabla^2 g(\mathbf{U})[\mathbf{D}, \mathbf{D}] - \langle \mathbf{U}\mathbf{\Omega}, \mathbf{D}\rangle = \nabla^2 g(\mathbf{U})[\mathbf{D}, \mathbf{D}]$$

for any $\mathbf{D} \in \mathcal{H}_{\mathbf{U}}\mathcal{M}$. Here, $\langle \mathbf{U}\mathbf{\Omega}, \mathbf{D}\rangle = \langle \mathbf{\Omega}, \mathbf{U}^\top \mathbf{D}\rangle = 0$ follows from the fact that $\mathbf{\Omega}$ is a skew-symmetric matrix and $\mathbf{D}^\top \mathbf{U} = \mathbf{U}^\top \mathbf{D}$.

## D.1  Determining critical points

By setting $\mathrm{grad}\, g(\mathbf{U}) = \mathbf{0}$, we get $\mathbf{X}\mathbf{U} = \mathbf{U}\mathbf{U}^\top \mathbf{U}$. Denote $\mathbf{U} = \mathbf{W}_u \mathbf{\Lambda}_u^{\frac{1}{2}} \mathbf{Q}^\top$ as an SVD of $\mathbf{U}$ with $\mathbf{W}_u \in \mathbb{R}^{N \times k}$, $\mathbf{\Lambda}_u \in \mathbb{R}^{k \times k}$ and $\mathbf{Q} \in \mathbb{R}^{k \times k}$. It follows from $\mathbf{X}\mathbf{U} = \mathbf{U}\mathbf{U}^\top \mathbf{U}$ that

$$\mathbf{X}\mathbf{W}_u \mathbf{\Lambda}_u^{\frac{1}{2}} \mathbf{Q}^\top = \mathbf{W}_u \mathbf{\Lambda}_u^{\frac{3}{2}} \mathbf{Q}^\top,$$

which further gives us

$$\mathbf{X}\mathbf{W}_u = \mathbf{W}_u \mathbf{\Lambda}_u.$$

For $i = 1, \dots, k$, denote $\boldsymbol{w}_{ui}$ and $\lambda_{ui}$ as the $i$-th column of $\mathbf{W}_u$ and $i$-th diagonal entry of $\mathbf{\Lambda}_u$, respectively. Then, we have

$$\mathbf{X}\boldsymbol{w}_{ui} = \lambda_{ui}\boldsymbol{w}_{ui},$$

which implies that $\lambda_{ui}$ is one of the eigenvalues of $\mathbf{X}$ and $\boldsymbol{w}_{ui}$ is the corresponding eigenvector. Therefore, any $\mathbf{U} \in \mathcal{U}$ is a critical point of $g(\mathbf{U})$ and we finish the proof of property (1).

## D.2  Strongly convexity in region $\mathcal{R}_1$

Recall that $\mathbf{U}^\star = \mathbf{W}_k \mathbf{\Lambda}_k^{\frac{1}{2}} \mathbf{Q}^\top$ with $\mathbf{\Lambda}_k = \mathrm{diag}([\lambda_1, \cdots, \lambda_k])$ containing the largest $k$ eigenvalues of $\mathbf{X}$. It follows from the Eckart-Young-Mirsky theorem [10] that any $\mathbf{U}^\star \in \mathcal{U}^\star$ is a global minimum of $g(\mathbf{U})$. Note that we can rewrite $\mathbf{X}$ as

$$\mathbf{X} = \mathbf{W}_k \mathbf{\Lambda}_k \mathbf{W}_k^\top + \mathbf{W}_k^\perp \mathbf{\Lambda}_k^\perp {\mathbf{W}_k^\perp}^\top = \mathbf{U}^\star {\mathbf{U}^\star}^\top + \mathbf{W}_k^\perp \mathbf{\Lambda}_k^\perp {\mathbf{W}_k^\perp}^\top, \tag{D.1}$$

where $\mathbf{W}_k^\perp \in \mathbb{R}^{N \times (r-k)}$ is a matrix that contains eigenvectors of $\mathbf{X}$ corresponding to eigenvalues in $\mathbf{\Lambda}_k^\perp = \mathrm{diag}([\lambda_{k+1}, \cdots, \lambda_r])$. For any $\mathbf{D} \in \mathbb{R}_*^{N \times k}$ that belongs to the horizontal space $\mathcal{H}_{\mathbf{U}^\star}\mathcal{M}$ at any $\mathbf{U}^\star \in \mathcal{U}^\star$, we have $\mathbf{D}^\top \mathbf{U}^\star = {\mathbf{U}^\star}^\top \mathbf{D}$, which implies that

$$\langle \mathbf{\Omega}, {\mathbf{U}^\star}^\top \mathbf{D}\rangle = 0,$$

since $\mathbf{\Omega}$ is a skew-symmetric matrix. Then, for $\forall\, \mathbf{D} \in \mathcal{H}_{\mathbf{U}^\star}\mathcal{M}$, we have

$$\begin{aligned}
\mathrm{hess}\, g(\mathbf{U}^\star)[\mathbf{D}, \mathbf{D}] &= \langle \nabla^2 g(\mathbf{U}^\star)[\mathbf{D}], \mathbf{D}\rangle - \langle \mathbf{U}^\star \mathbf{\Omega}, \mathbf{D}\rangle \\
&\overset{\text{①}}{=} \langle \nabla^2 g(\mathbf{U}^\star)[\mathbf{D}], \mathbf{D}\rangle \\
&= \langle (\mathbf{U}^\star \mathbf{D}^\top + \mathbf{D}{\mathbf{U}^\star}^\top)\mathbf{U}^\star + (\mathbf{U}^\star {\mathbf{U}^\star}^\top - \mathbf{X})\mathbf{D}, \mathbf{D}\rangle \\
&= \langle \mathbf{W}_k \mathbf{\Lambda}_k \mathbf{W}_k^\top, \mathbf{D}\mathbf{D}^\top\rangle + \langle \mathbf{Q}\mathbf{\Lambda}_k \mathbf{Q}^\top, \mathbf{D}^\top \mathbf{D}\rangle - \langle \mathbf{W}_k^\perp \mathbf{\Lambda}_k^\perp {\mathbf{W}_k^\perp}^\top, \mathbf{D}\mathbf{D}^\top\rangle \\
&\overset{\text{②}}{\geq} \lambda_k \|\mathbf{D}\|_F^2 + \lambda_k \|\mathbf{D}\|_F^2 - \lambda_{k+1}\|\mathbf{D}\|_F^2 \\
&\overset{\text{③}}{\geq} 1.91\lambda_k \|\mathbf{D}\|_F^2.
\end{aligned}$$

Here, ① follows from $\langle \mathbf{U}^\star \mathbf{\Omega}, \mathbf{D} \rangle = \langle \mathbf{\Omega}, \mathbf{U}^{\star \top} \mathbf{D} \rangle = 0$, ② follows from [11, Lemma 7], and ③ follows from the assumption $\lambda_{k+1} \le \frac{1}{12}\lambda_k$. Then, we have

$$\lambda_{\min}(\text{hess } g(\mathbf{U}^\star)) \ge 1.91\lambda_k > 0, \tag{D.2}$$

which also implies that any $\mathbf{U}^\star \in \mathcal{U}^\star$ is a strict local minimum of $g(\mathbf{U})$.

Next, we characterize the strong convexity in region $\mathcal{R}_1$. Note that for $\forall\ x_1,\ x_2\ \in\ \mathbb{R}$, we have $x_1 - x_2 \ge -|x_1 - x_2|$, i.e., $x_1 \ge x_2 - |x_1 - x_2|$, which implies that

$$\text{hess } g(\mathbf{U})[\mathbf{D}, \mathbf{D}] \ge \text{hess } g(\mathbf{U}^\star)[\mathbf{D}, \mathbf{D}] - |\text{hess } g(\mathbf{U})[\mathbf{D}, \mathbf{D}] - \text{hess } g(\mathbf{U}^\star)[\mathbf{D}, \mathbf{D}]|, \tag{D.3}$$

where $\mathbf{D}$ belongs to the horizontal space $\mathcal{H}_\mathbf{U}\mathcal{M}$ at any $\mathbf{U} \in \mathcal{R}_1$, i.e., $\mathbf{U}^\top \mathbf{D} = \mathbf{D}^\top \mathbf{U}$. For notational simplicity, we denote $\mathbf{U}^\star \mathbf{P}^\star$ with $\mathbf{P}^\star = \arg\min_{\mathbf{P} \in \mathcal{O}_k} \|\mathbf{U} - \mathbf{U}^\star \mathbf{P}\|_F$ as $\mathbf{U}^\star$. In the rest of this section, we bound the two terms in the right hand side of (D.3) in sequence.

**Term 1:** Note that hess $g(\mathbf{U}^\star)[\mathbf{D}]$ is the projection of $\nabla^2 g(\mathbf{U}^\star)[\mathbf{D}]$ onto the horizontal space $\mathcal{H}_\mathbf{U}\mathcal{M}$, namely, hess $g(\mathbf{U}^\star)[\mathbf{D}] = \nabla^2 g(\mathbf{U}^\star)[\mathbf{D}] - \mathbf{U}\mathbf{\Omega}$ with $\mathbf{\Omega}$ being a skew-symmetric matrix that solves the following Sylvester equation

$$\mathbf{\Omega}\mathbf{U}^\top\mathbf{U} + \mathbf{U}^\top\mathbf{U}\mathbf{\Omega} = \mathbf{U}^\top\nabla^2 g(\mathbf{U}^\star)[\mathbf{D}] - \nabla^2 g(\mathbf{U}^\star)[\mathbf{D}]^\top\mathbf{U}. \tag{D.4}$$

Then, we have

$$\begin{aligned}
\text{hess } g(\mathbf{U}^\star)[\mathbf{D}, \mathbf{D}] &= \langle \nabla^2 g(\mathbf{U}^\star)[\mathbf{D}], \mathbf{D} \rangle - \langle \mathbf{U}\mathbf{\Omega}, \mathbf{D} \rangle \\
&= \langle \nabla^2 g(\mathbf{U}^\star)[\mathbf{D}], \mathbf{D} \rangle,
\end{aligned} \tag{D.5}$$

where the second line follows from $\langle \mathbf{U}\mathbf{\Omega}, \mathbf{D} \rangle = \langle \mathbf{\Omega}, \mathbf{U}^\top\mathbf{D} \rangle$, $\mathbf{U}^\top\mathbf{D} = \mathbf{D}^\top\mathbf{U}$ and $\mathbf{\Omega} + \mathbf{\Omega}^\top = \mathbf{0}$. Defining $\mathbf{E}_u \triangleq \mathbf{U} - \mathbf{U}^\star$, together with $\mathbf{U}^\top\mathbf{D} = \mathbf{D}^\top\mathbf{U}$, we obtain

$$(\mathbf{U}^\star + \mathbf{E}_u)^\top\mathbf{D} = \mathbf{D}^\top(\mathbf{U}^\star + \mathbf{E}_u),$$

which further gives us

$$\mathbf{D}^\top\mathbf{U}^\star = \mathbf{U}^{\star\top}\mathbf{D} + \mathbf{E}_u^\top\mathbf{D} - \mathbf{D}^\top\mathbf{E}_u. \tag{D.6}$$

By combining (D.5) and (D.6), we can bound the first term with

$$\begin{aligned}
&\text{hess } g(\mathbf{U}^\star)[\mathbf{D}, \mathbf{D}] = \langle \nabla^2 g(\mathbf{U}^\star)[\mathbf{D}], \mathbf{D} \rangle \\
=&\langle (\mathbf{U}^\star\mathbf{D}^\top + \mathbf{D}\mathbf{U}^{\star\top})\mathbf{U}^\star + (\mathbf{U}^\star\mathbf{U}^{\star\top} - \mathbf{X})\mathbf{D}, \mathbf{D} \rangle \\
=&\langle \mathbf{D}^\top\mathbf{U}^\star, \mathbf{U}^{\star\top}\mathbf{D} \rangle + \langle \mathbf{U}^{\star\top}\mathbf{U}^\star, \mathbf{D}^\top\mathbf{D} \rangle - \langle \mathbf{W}_k^\perp \mathbf{\Lambda}_k^\perp \mathbf{W}_k^{\perp\top}, \mathbf{D}\mathbf{D}^\top \rangle \\
=&\langle \mathbf{U}^{\star\top}\mathbf{D} + \mathbf{E}_u^\top\mathbf{D} - \mathbf{D}^\top\mathbf{E}_u, \mathbf{U}^{\star\top}\mathbf{D} \rangle + \langle \mathbf{Q}\mathbf{\Lambda}_k\mathbf{Q}^\top, \mathbf{D}^\top\mathbf{D} \rangle - \langle \mathbf{W}_k^\perp \mathbf{\Lambda}_k^\perp \mathbf{W}_k^{\perp\top}, \mathbf{D}\mathbf{D}^\top \rangle \\
=&\langle \mathbf{U}^\star\mathbf{U}^{\star\top}, \mathbf{D}\mathbf{D}^\top \rangle + \langle \mathbf{E}_u^\top\mathbf{D}, \mathbf{U}^{\star\top}\mathbf{D} \rangle - \langle \mathbf{D}^\top\mathbf{E}_u, \mathbf{U}^{\star\top}\mathbf{D} \rangle + \langle \mathbf{Q}\mathbf{\Lambda}_k\mathbf{Q}^\top, \mathbf{D}^\top\mathbf{D} \rangle - \langle \mathbf{W}_k^\perp \mathbf{\Lambda}_k^\perp \mathbf{W}_k^{\perp\top}, \mathbf{D}\mathbf{D}^\top \rangle \\
\ge& \lambda_k\|\mathbf{D}\|_F^2 - 0.2\lambda_k\|\mathbf{D}\|_F^2 - 0.2\lambda_k\|\mathbf{D}\|_F^2 + \lambda_k\|\mathbf{D}\|_F^2 - \frac{1}{12}\lambda_k\|\mathbf{D}\|_F^2 \\
\ge& 1.51\lambda_k\|\mathbf{D}\|_F^2,
\end{aligned}$$

where the first inequality follows from [11, Lemma 7], the Matrix Hölder Inequality [12], the assumption $\lambda_{k+1} \le \frac{1}{12}\lambda_k$, and the following two inequalities

$$\begin{aligned}
\langle \mathbf{E}_u^\top\mathbf{D}, \mathbf{U}^{\star\top}\mathbf{D} \rangle &\ge -\|\mathbf{E}_u^\top\mathbf{D}\|_F\|\mathbf{U}^{\star\top}\mathbf{D}\|_F \ge -\|\mathbf{E}_u\|_F\|\mathbf{U}^\star\|_2\|\mathbf{D}\|_F^2 \\
&\ge -0.2\kappa^{-1}\sqrt{\lambda_k}\sqrt{\lambda_1}\|\mathbf{D}\|_F^2 = -0.2\lambda_k\|\mathbf{D}\|_F^2,
\end{aligned}$$

$$\begin{aligned}
\langle \mathbf{D}^\top\mathbf{E}_u, \mathbf{U}^{\star\top}\mathbf{D} \rangle &\le \|\mathbf{D}^\top\mathbf{E}_u\|_F\|\mathbf{U}^{\star\top}\mathbf{D}\|_F \le \|\mathbf{E}_u\|_F\|\mathbf{U}^\star\|_2\|\mathbf{D}\|_F^2 \\
&\le 0.2\kappa^{-1}\sqrt{\lambda_k}\sqrt{\lambda_1}\|\mathbf{D}\|_F^2 = 0.2\lambda_k\|\mathbf{D}\|_F^2.
\end{aligned}$$

**Term 2:** By plugging hess $g(\mathbf{U})[\mathbf{D}, \mathbf{D}] = \langle (\mathbf{U}\mathbf{D}^\top + \mathbf{D}\mathbf{U}^\top)\mathbf{U} + (\mathbf{U}\mathbf{U}^\top - \mathbf{X})\mathbf{D}, \mathbf{D} \rangle$ and hess $g(\mathbf{U}^\star)[\mathbf{D}, \mathbf{D}] = \langle (\mathbf{U}^\star\mathbf{D}^\top + \mathbf{D}\mathbf{U}^{\star\top})\mathbf{U}^\star + (\mathbf{U}^\star\mathbf{U}^{\star\top} - \mathbf{X})\mathbf{D}, \mathbf{D} \rangle$ into the second term, we

obtain

$$|\operatorname{hess} g(\mathbf{U})[\mathbf{D}, \mathbf{D}] - \operatorname{hess} g(\mathbf{U}^\star)[\mathbf{D}, \mathbf{D}]|$$

$$=|2\langle \mathbf{U}\mathbf{U}^\top - \mathbf{U}^\star \mathbf{U}^{\star\top}, \mathbf{D}\mathbf{D}^\top\rangle - \langle \mathbf{U}^\star \mathbf{E}_u^\top, \mathbf{D}\mathbf{D}^\top\rangle + \langle \mathbf{D}^\top \mathbf{E}_u, \mathbf{U}^{\star\top}\mathbf{D}\rangle + \langle \mathbf{U}^\top \mathbf{U} - \mathbf{U}^{\star\top}\mathbf{U}^\star\rangle|$$

$$=|3\langle \mathbf{U}^{\star\top}\mathbf{D}, \mathbf{E}_u^\top \mathbf{D}\rangle + 2\langle \mathbf{E}_u^\top \mathbf{D}, \mathbf{E}_u^\top \mathbf{D}\rangle + \langle \mathbf{D}^\top \mathbf{E}_u, \mathbf{U}^{\star\top}\mathbf{D}\rangle + 2\langle \mathbf{D}\mathbf{E}_u^\top, \mathbf{D}\mathbf{U}^{\star\top}\rangle + \langle \mathbf{D}\mathbf{E}_u^\top, \mathbf{D}\mathbf{E}_u^\top\rangle|$$

$$\leq 6\|\mathbf{U}^\star\|_2\|\mathbf{E}_u\|_F\|\mathbf{D}\|_F^2 + 3\|\mathbf{E}_u\|_F^2\|\mathbf{D}\|_F^2$$

$$\leq 1.2\lambda_k\|\mathbf{D}\|_F^2 + 0.12\kappa^{-2}\lambda\|\mathbf{D}\|_F^2$$

$$\leq 1.32\lambda_k\|\mathbf{D}\|_F^2,$$

where the first inequality follows from the Triangle Inequality and the Matrix Hölder Inequality [12], and the last two inequalities follow from $\|\mathbf{U}^\star\|_2 = \sqrt{\lambda_1}$ and $\|\mathbf{E}_u\|_F \leq 0.2\kappa^{-1}\sqrt{\lambda_k}$ with $\kappa \geq 1$.

As a consequence, we have

$$\operatorname{hess} g(\mathbf{U})[\mathbf{D}, \mathbf{D}] \geq \operatorname{hess} g(\mathbf{U}^\star)[\mathbf{D}, \mathbf{D}] - |\operatorname{hess} g(\mathbf{U})[\mathbf{D}, \mathbf{D}] - \operatorname{hess} g(\mathbf{U}^\star)[\mathbf{D}, \mathbf{D}]|$$

$$\geq 0.19\lambda_k\|\mathbf{D}\|_F^2,$$

which implies that

$$\lambda_{\min}(\operatorname{hess} g(\mathbf{U})) \geq 0.19\lambda_k$$

holds for any $\mathbf{U} \in \mathcal{R}_1$. Thus, we finish the proof of property (2).

### D.3 Negative curvature in region $\mathcal{R}_2'$

For any $\mathbf{U}_s^\star \in \mathcal{U}_s^\star$, let $\mathbf{U}_s^\star = \mathbf{W}_s \mathbf{\Lambda}_s^{\frac{1}{2}} \mathbf{Q}^\top$ be an SVD of $\mathbf{U}_s^\star$ with $\mathbf{W}_s \in \mathbb{R}^{N\times k}$, $\mathbf{\Lambda}_s \in \mathbb{R}^{k\times k}$ and $\mathbf{Q} \in \mathcal{O}_k$. According to the definition of $\mathcal{U}_s^\star$, $\mathbf{\Lambda}_s \in \mathbb{R}^{k\times k}$ contains any $k$ non-zero eigenvalues of $\mathbf{X}$ except the largest $k$ eigenvalues. Denote $\mathbf{\Lambda}_s = \operatorname{diag}([\lambda_{s1}, \cdots, \lambda_{sk}])$ with $\lambda_{s1} \geq \cdots \geq \lambda_{sk} > 0$, we have $\lambda_{sk} \leq \lambda_{k+1}$. Let $\boldsymbol{q}_k$ denote the $k$-th column of $\mathbf{Q}$. $\boldsymbol{w}^\star \in \mathbb{R}^N$ is one column chosen from $\mathbf{W}_k$ satisfying $\boldsymbol{w}^{\star\top}\mathbf{W}_s = \mathbf{0}$. Then, we show that the function $g(\mathbf{U})$ at $\mathbf{U}_s^\star$ has directional negative curvature along the direction $\mathbf{D} = \boldsymbol{w}^\star \boldsymbol{q}_k^\top$. Note that

$$\mathbf{D}^\top \mathbf{U}_s^\star = \boldsymbol{q}_k \boldsymbol{w}^{\star\top}\mathbf{U}_s^\star = \mathbf{0},$$

$$\mathbf{U}_s^{\star\top}\mathbf{D} = \mathbf{U}_s^{\star\top}\boldsymbol{w}^\star \boldsymbol{q}_k^\top = \mathbf{0},$$

which verifies that this direction $\mathbf{D} = \boldsymbol{w}^\star \boldsymbol{q}_k^\top$ belongs to the horizontal space $\mathcal{H}_{\mathbf{U}_s^\star}\mathcal{M}$ at $\mathbf{U}_s^\star$. It can be seen that

$$\operatorname{hess} g(\mathbf{U}_s^\star)[\mathbf{D}, \mathbf{D}] = \langle (\mathbf{U}_s^\star\mathbf{D}^\top + \mathbf{D}\mathbf{U}_s^{\star\top})\mathbf{U}_s^\star + (\mathbf{U}_s^\star\mathbf{U}_s^{\star\top} - \mathbf{X})\mathbf{D}, \mathbf{D}\rangle$$

$$= \langle \mathbf{U}_s^{\star\top}\mathbf{U}_s^\star, \mathbf{D}^\top \mathbf{D}\rangle - \langle \mathbf{W}_s^\perp \mathbf{\Lambda}_s^\perp \mathbf{W}_s^{\perp\top}, \mathbf{D}\mathbf{D}^\top\rangle$$

$$= \langle \mathbf{Q}\mathbf{\Lambda}_s\mathbf{Q}^\top, \boldsymbol{q}_k\boldsymbol{q}_k^\top\rangle - \langle \mathbf{W}_s^\perp \mathbf{\Lambda}_s^\perp \mathbf{W}_s^{\perp\top}, \boldsymbol{w}^\star \boldsymbol{w}^{\star\top}\rangle$$

$$\leq \lambda_{sk} - \lambda_k \leq -0.91\lambda_k = -0.91\lambda_k\|\mathbf{D}\|_F^2,$$

where $\mathbf{W}_s^\perp \in \mathbb{R}^{N\times(r-k)}$ is a matrix that contains eigenvectors of $\mathbf{X}$ corresponding to eigenvalues in $\mathbf{\Lambda}_s^\perp$, i.e., eigenvalues of $\mathbf{X}$ not contained in $\mathbf{\Lambda}_s$. The first inequality follows since $\boldsymbol{w}^\star$ is a column of both $\mathbf{W}_s^\perp$ and $\mathbf{W}_k$. The second inequality follows from $\lambda_{sk} \leq \lambda_{k+1} \leq \frac{1}{12}\lambda_k$. Therefore, we have

$$\lambda_{\min}(\operatorname{hess} g(\mathbf{U}_s^\star)) \leq -0.91\lambda_k.$$

Next, we show that the function $g(\mathbf{U})$ has directional negative curvature for any $\mathbf{U} \in \mathcal{R}_2'$ along the direction

$$\mathbf{D} = \mathbf{U} - \mathbf{U}^\star \mathbf{P}^\star \text{ with } \mathbf{P}^\star = \arg\min_{\mathbf{P}\in\mathcal{O}_k}\|\mathbf{U} - \mathbf{U}^\star \mathbf{P}\|_F.$$

For notational simplicity, we still denote $\mathbf{U}^\star \mathbf{P}^\star$ as $\mathbf{U}^\star$, i.e., $\mathbf{D} = \mathbf{U} - \mathbf{U}^\star$. First, we need to verify that this direction belongs to the horizontal space $\mathcal{H}_{\mathbf{U}}\mathcal{M}$ at $\mathbf{U}$. As is shown in [13, proof of Lemma 6], $\mathbf{U}^\top \mathbf{U}^\star$ is a symmetric PSD matrix. Then, we have

$$\mathbf{D}^\top \mathbf{U} = \mathbf{U}^\top \mathbf{U} - \mathbf{U}^{\star\top}\mathbf{U} = \mathbf{U}^\top \mathbf{U} - \mathbf{U}^\top \mathbf{U}^\star = \mathbf{U}^\top \mathbf{D},$$

which implies that $\mathbf{D} \in \mathcal{H}_{\mathbf{U}}\mathcal{M}$.

Note that minimizing $g(\mathbf{U})$ is equivalent to the following minimization problem

$$\min_{\mathbf{U}\in\mathbb{R}_*^{N\times k}} \frac{1}{2}\|\mathbf{U}\mathbf{U}^\top - \mathbf{U}^\star\mathbf{U}^{\star\top}\|_F^2 - \langle\mathbf{U}\mathbf{U}^\top, \mathbf{W}_k^\perp\mathbf{\Lambda}_k^\perp\mathbf{W}_k^{\perp\top}\rangle.$$

Define two functions $g_1(\mathbf{U})$ and $g_2(\mathbf{U})$ as

$$g_1(\mathbf{U}) \triangleq \frac{1}{2}\|\mathbf{U}\mathbf{U}^\top - \mathbf{U}^\star\mathbf{U}^{\star\top}\|_F^2 - \langle\mathbf{U}\mathbf{U}^\top, \mathbf{W}_k^\perp\mathbf{\Lambda}_k^\perp\mathbf{W}_k^{\perp\top}\rangle,$$

$$g_2(\mathbf{U}) \triangleq -\langle\mathbf{U}\mathbf{U}^\top, \mathbf{W}_k^\perp\mathbf{\Lambda}_k^\perp\mathbf{W}_k^{\perp\top}\rangle.$$

Then, we have

$$\nabla g_2(\mathbf{U}) = -2\mathbf{W}_k^\perp\mathbf{\Lambda}_k^\perp\mathbf{W}_k^{\perp\top}\mathbf{U},$$

$$\nabla^2 g_2(\mathbf{U})[\mathbf{D},\mathbf{D}] = -2\langle\mathbf{W}_k^\perp\mathbf{\Lambda}_k^\perp\mathbf{W}_k^{\perp\top}, \mathbf{D}\mathbf{D}^\top\rangle.$$

Together with [13, Lemma 7], we get

$$\begin{aligned}
2\,\mathrm{hess}\,g(\mathbf{U})[\mathbf{D},\mathbf{D}] &= 2\nabla^2 g(\mathbf{U})[\mathbf{D},\mathbf{D}] = \nabla^2 g_1(\mathbf{U})[\mathbf{D},\mathbf{D}] \\
&= \|\mathbf{D}\mathbf{D}^\top\|_F^2 - 3\|\mathbf{U}\mathbf{U}^\top - \mathbf{U}^\star\mathbf{U}^{\star\top}\|_F^2 + 4\langle\nabla g_1(\mathbf{U}),\mathbf{D}\rangle + \nabla^2 g_2(\mathbf{U})[\mathbf{D},\mathbf{D}] - 4\langle\nabla g_2(\mathbf{U}),\mathbf{D}\rangle,
\end{aligned} \tag{D.7}$$

where the first equality follows from $\langle\mathbf{U}\mathbf{\Omega},\mathbf{D}\rangle = 0$, similar to Appendix D.3.

Note that the first two terms in (D.7) can be bounded with

$$\begin{aligned}
&\|\mathbf{D}\mathbf{D}^\top\|_F^2 - 3\|\mathbf{U}\mathbf{U}^\top - \mathbf{U}^\star\mathbf{U}^{\star\top}\|_F^2 \\
&\leq -\|\mathbf{U}\mathbf{U}^\top - \mathbf{U}^\star\mathbf{U}^{\star\top}\|_F^2 \\
&\leq -2(\sqrt{2}-1)\lambda_k\|\mathbf{D}\|_F^2 \\
&\leq -0.82\lambda_k\|\mathbf{D}\|_F^2
\end{aligned} \tag{D.8}$$

by using Lemma 6 in [13].

Note that

$$\|\mathbf{D}\|_F = \|\mathbf{U}-\mathbf{U}^\star\|_F \geq \|\mathrm{diag}([\sigma_1(\mathbf{U})-\sqrt{\lambda_1},\cdots,\sigma_k(\mathbf{U})-\sqrt{\lambda_k}])\|_F \geq \sigma_k(\mathbf{U})-\sqrt{\lambda_k} \geq \frac{1}{2}\sqrt{\lambda_k},$$

where the first inequality follows from [3, Theorem 5], and the last inequality follows from $\sigma_k(\mathbf{U}) \leq \frac{1}{2}\sqrt{\lambda_k}$. Then, the third term in (D.7) can be bounded with

$$\begin{aligned}
\langle\nabla g_1(\mathbf{U}),\mathbf{D}\rangle &\leq \|\nabla g_1(\mathbf{U})\|_F\|\mathbf{D}\|_F = 2\|\mathrm{grad}\,g(\mathbf{U})\|_F\|\mathbf{D}\|_F \\
&\leq \frac{1}{40}\lambda_k^{\frac{3}{2}}\|\mathbf{D}\|_F = \frac{1}{20}\lambda_k\|\mathbf{D}\|_F\frac{1}{2}\sqrt{\lambda_k} \leq \frac{1}{20}\lambda_k\|\mathbf{D}\|_F^2.
\end{aligned} \tag{D.9}$$

Next, we bound the last two terms in (D.7) with

$$\begin{aligned}
&\nabla^2 g_2(\mathbf{U})[\mathbf{D},\mathbf{D}] - 4\langle\nabla g_2(\mathbf{U}),\mathbf{D}\rangle \\
&= -2\langle\mathbf{W}_k^\perp\mathbf{\Lambda}_k^\perp\mathbf{W}_k^{\perp\top}, \mathbf{D}\mathbf{D}^\top\rangle + 8\langle\mathbf{W}_k^\perp\mathbf{\Lambda}_k^\perp\mathbf{W}_k^{\perp\top}\mathbf{U},\mathbf{D}\rangle \\
&= 8\langle\mathbf{W}_k^\perp\mathbf{\Lambda}_k^\perp\mathbf{W}_k^{\perp\top}\mathbf{U},\mathbf{U}-\mathbf{U}^\star\rangle - 2\langle\mathbf{W}_k^\perp\mathbf{\Lambda}_k^\perp\mathbf{W}_k^{\perp\top}, \mathbf{U}\mathbf{U}^\top - \mathbf{U}^\star\mathbf{U}^\top - \mathbf{U}\mathbf{U}^{\star\top} + \mathbf{U}^\star\mathbf{U}^{\star\top}\rangle \\
&\overset{\textcircled{1}}{=} 6\langle\mathbf{W}_k^\perp\mathbf{\Lambda}_k^\perp\mathbf{W}_k^{\perp\top}, \mathbf{U}\mathbf{U}^\top\rangle = 6\langle\mathbf{\Lambda}_k^\perp, \mathbf{W}_k^{\perp\top}\mathbf{U}\mathbf{U}^\top\mathbf{W}_k^\perp\rangle \\
&\overset{\textcircled{2}}{\leq} 6\lambda_{k+1}\|\mathbf{W}_k^{\perp\top}\mathbf{U}\|_F^2 \overset{\textcircled{3}}{=} 6\lambda_{k+1}\|\mathbf{W}_k^{\perp\top}(\mathbf{U}-\mathbf{U}^\star)\|_F^2 \\
&\leq \frac{1}{2}\lambda_k\|\mathbf{D}\|_F^2,
\end{aligned} \tag{D.10}$$

where $\textcircled{1}$ and $\textcircled{3}$ follow from $\mathbf{W}_k^{\perp\top}\mathbf{U}^\star = \mathbf{0}$, and $\textcircled{2}$ follows from [11, Lemma 7].

By plugging inequalities (D.8), (D.9) and (D.10) into (D.7), we obtain

$$\mathrm{hess}\,g(\mathbf{U})[\mathbf{D},\mathbf{D}] \leq -0.41\lambda_k\|\mathbf{D}\|_F^2 + 0.1\lambda_k\|\mathbf{D}\|_F^2 + 0.25\lambda_k\|\mathbf{D}\|_F^2 = -0.06\lambda_k\|\mathbf{D}\|_F^2,$$

which implies that

$$\lambda_{\min}(\mathrm{hess}\,g(\mathbf{U})) \leq -0.06\lambda_k$$

holds for all $\mathbf{U} \in \mathcal{R}_2'$, and we finish the proof of property (3).

## D.4 Large gradient in regions $\mathcal{R}_2''$, $\mathcal{R}_3'$ and $\mathcal{R}_3''$

It is easy to see that the first inequality in property (4) is true due to the definition of $\mathcal{R}_2''$. In this section, we mainly focus on showing the gradient is large in regions $\mathcal{R}_3'$ and $\mathcal{R}_3''$.

### D.4.1 Large gradient in region $\mathcal{R}_3'$

To show $\|\operatorname{grad} g(\mathbf{U})\|_F$ is large for any $\mathbf{U} \in \mathcal{R}_3'$, we rewrite $\mathbf{U}$ as

$$\mathbf{U} = \mathbf{W}_k \widetilde{\mathbf{\Lambda}}_u^{\frac{1}{2}} \widetilde{\mathbf{Q}}_u^\top + \widetilde{\mathbf{E}}_u, \tag{D.11}$$

where $\mathbf{W}_k \in \mathbb{R}^{N \times k}$ contains the $k$ eigenvectors of $\mathbf{X}$ associated with the $k$ largest eigenvalues of $\mathbf{X}$, $\widetilde{\mathbf{\Lambda}}_u \in \mathbb{R}^{k \times k}$ is a diagonal matrix, $\widetilde{\mathbf{Q}}_u \in \mathcal{O}_k$ is an orthogonal matrix, and $\widetilde{\mathbf{E}}_u^\top \mathbf{W}_k = \mathbf{0}$. Note that $\mathbf{W}_k \widetilde{\mathbf{\Lambda}}_u^{\frac{1}{2}} \widetilde{\mathbf{Q}}_u^\top$ can be viewed as a compact SVD form of the projection of $\mathbf{U}$ onto the column space of $\mathbf{W}_k$. Plugging (D.11) and (D.1) into $\|\operatorname{grad} g(\mathbf{U})\|_F^2$ gives

$$
\begin{aligned}
&\|\operatorname{grad} g(\mathbf{U})\|_F^2 = \|(\mathbf{U}\mathbf{U}^\top - \mathbf{X})\mathbf{U}\|_F^2 \\
&= \|\mathbf{W}_k \widetilde{\mathbf{\Lambda}}_u^{\frac{1}{2}} (\widetilde{\mathbf{\Lambda}}_u - \mathbf{\Lambda}_k)\widetilde{\mathbf{Q}}_u^\top + \mathbf{W}_k \widetilde{\mathbf{\Lambda}}_u^{\frac{1}{2}}\widetilde{\mathbf{Q}}_u^\top \widetilde{\mathbf{E}}_u^\top \widetilde{\mathbf{E}}_u + \widetilde{\mathbf{E}}_u \widetilde{\mathbf{Q}}_u \widetilde{\mathbf{\Lambda}}_u \widetilde{\mathbf{Q}}_u^\top + \widetilde{\mathbf{E}}_u \widetilde{\mathbf{E}}_u^\top \widetilde{\mathbf{E}}_u - \mathbf{W}_k^\perp \mathbf{\Lambda}_k^\perp \mathbf{W}_k^{\perp\top} \widetilde{\mathbf{E}}_u\|_F^2 \\
&= \|\widetilde{\mathbf{E}}_u \widetilde{\mathbf{Q}}_u \widetilde{\mathbf{\Lambda}}_u \widetilde{\mathbf{Q}}_u^\top + \widetilde{\mathbf{E}}_u \widetilde{\mathbf{E}}_u^\top \widetilde{\mathbf{E}}_u - \mathbf{W}_k^\perp \mathbf{\Lambda}_k^\perp \mathbf{W}_k^{\perp\top} \widetilde{\mathbf{E}}_u\|_F^2 + \|\mathbf{W}_k \widetilde{\mathbf{\Lambda}}_u^{\frac{1}{2}} (\widetilde{\mathbf{\Lambda}}_u - \mathbf{\Lambda}_k)\widetilde{\mathbf{Q}}_u^\top + \mathbf{W}_k \widetilde{\mathbf{\Lambda}}_u^{\frac{1}{2}}\widetilde{\mathbf{Q}}_u^\top \widetilde{\mathbf{E}}_u^\top \widetilde{\mathbf{E}}_u\|_F^2,
\end{aligned} \tag{D.12}
$$

where the last equality follows from $\widetilde{\mathbf{E}}_u^\top \mathbf{W}_k = \mathbf{0}$. Next, we show at least one of the above two terms is large for any $\mathbf{U} \in \mathcal{R}_3'$ by considering the following two cases.

**Case 1:** $\|\widetilde{\mathbf{E}}_u\|_F \geq 0.1\kappa^{-1}\sqrt{\lambda_k}$. The square root of the first term in (D.12) can be bounded with

$$
\begin{aligned}
&\|\widetilde{\mathbf{E}}_u \widetilde{\mathbf{Q}}_u \widetilde{\mathbf{\Lambda}}_u \widetilde{\mathbf{Q}}_u^\top + \widetilde{\mathbf{E}}_u \widetilde{\mathbf{E}}_u^\top \widetilde{\mathbf{E}}_u - \mathbf{W}_k^\perp \mathbf{\Lambda}_k^\perp \mathbf{W}_k^{\perp\top} \widetilde{\mathbf{E}}_u\|_F \\
&\geq \|\widetilde{\mathbf{E}}_u (\widetilde{\mathbf{Q}}_u \widetilde{\mathbf{\Lambda}}_u \widetilde{\mathbf{Q}}_u^\top + \widetilde{\mathbf{E}}_u^\top \widetilde{\mathbf{E}}_u)\|_F - \|\mathbf{W}_k^\perp \mathbf{\Lambda}_k^\perp \mathbf{W}_k^{\perp\top} \widetilde{\mathbf{E}}_u\|_F \\
&\overset{\text{①}}{\geq} \sigma_k(\mathbf{U}^\top \mathbf{U})\|\widetilde{\mathbf{E}}_u\|_F - \lambda_{k+1}\|\widetilde{\mathbf{E}}_u\|_F \\
&\overset{\text{②}}{>} \frac{1}{6}\lambda_k \|\widetilde{\mathbf{E}}_u\|_F \geq \frac{1}{60}\kappa^{-1}\lambda_k^{\frac{3}{2}}
\end{aligned} \tag{D.13}
$$

where ① follows from $\mathbf{U}^\top \mathbf{U} = \widetilde{\mathbf{Q}}_u \widetilde{\mathbf{\Lambda}}_u \widetilde{\mathbf{Q}}_u^\top + \widetilde{\mathbf{E}}_u^\top \widetilde{\mathbf{E}}_u$ and [11, Corollary 2], and ② follows from $\sigma_k(\mathbf{U}) > \frac{1}{2}\sqrt{\lambda_k}$ and the assumption $\lambda_{k+1} \leq \frac{1}{12}\lambda_k$.

**Case 2:** $\|\widetilde{\mathbf{E}}_u\|_F < 0.1\kappa^{-1}\sqrt{\lambda_k}$. Denote $\widetilde{\lambda}_{ui}$ as the $i$-th diagonal entry of $\widetilde{\mathbf{\Lambda}}_u$ with $\widetilde{\lambda}_{u1} \geq \cdots \geq \widetilde{\lambda}_{uk}$, i.e., $\sqrt{\widetilde{\lambda}_{ui}}$ is the $i$-th singular value of $\mathbf{W}_k \widetilde{\mathbf{\Lambda}}_u^{\frac{1}{2}} \widetilde{\mathbf{Q}}_u^\top$. By using Weyl's inequality for the perturbation of singular values [14] and (D.11), we get

$$\sigma_k(\mathbf{U}) - \sqrt{\widetilde{\lambda}_{uk}} \leq \|\widetilde{\mathbf{E}}_u\|_2 \leq \|\widetilde{\mathbf{E}}_u\|_F,$$

which further gives

$$\sqrt{\widetilde{\lambda}_{uk}} \geq \sigma_k(\mathbf{U}) - \|\widetilde{\mathbf{E}}_u\|_F > (0.5 - 0.1\kappa^{-1})\sqrt{\lambda_k}.$$

To bound the second term in (D.12), we still need a lower bound on $\|\widetilde{\mathbf{\Lambda}}_u - \mathbf{\Lambda}_k\|_F$. Recall that $\mathbf{Q} \in \mathcal{O}_k$ contains the right singular vectors of $\mathbf{U}^\star$. According to the definition of $\mathcal{R}_3'$, we have

$$
\begin{aligned}
0.2\kappa^{-1}\sqrt{\lambda_k} &< \min_{\mathbf{P} \in \mathcal{O}_k} \|\mathbf{U} - \mathbf{U}^\star \mathbf{P}\|_F \leq \|\mathbf{U} - \mathbf{U}^\star \mathbf{Q}\widetilde{\mathbf{Q}}_u^\top\|_F \\
&= \|\mathbf{W}_k \widetilde{\mathbf{\Lambda}}_u^{\frac{1}{2}} \widetilde{\mathbf{Q}}_u^\top + \widetilde{\mathbf{E}}_u - \mathbf{W}_k \mathbf{\Lambda}_k^{\frac{1}{2}} \widetilde{\mathbf{Q}}_u^\top\|_F \leq \|\mathbf{W}_k (\widetilde{\mathbf{\Lambda}}_u^{\frac{1}{2}} - \mathbf{\Lambda}_k^{\frac{1}{2}})\widetilde{\mathbf{Q}}_u^\top\|_F + \|\widetilde{\mathbf{E}}_u\|_F \\
&= \|\widetilde{\mathbf{\Lambda}}_u^{\frac{1}{2}} - \mathbf{\Lambda}_k^{\frac{1}{2}}\|_F + \|\widetilde{\mathbf{E}}_u\|_F,
\end{aligned}
$$

which implies

$$\|\widetilde{\mathbf{\Lambda}}_u^{\frac{1}{2}} - \mathbf{\Lambda}_k^{\frac{1}{2}}\|_F > 0.2\kappa^{-1}\sqrt{\lambda_k} - 0.1\kappa^{-1}\sqrt{\lambda_k} = 0.1\kappa^{-1}\sqrt{\lambda_k}.$$

Then, we can bound $\|\widetilde{\mathbf{\Lambda}}_u - \mathbf{\Lambda}_k\|_F$ with

$$\|\widetilde{\mathbf{\Lambda}}_u - \mathbf{\Lambda}_k\|_F = \sqrt{\sum_{i=1}^k (\widetilde{\lambda}_{ui} - \lambda_i)^2} = \sqrt{\sum_{i=1}^k (\sqrt{\widetilde{\lambda}_{ui}} - \sqrt{\lambda_i})^2 (\sqrt{\widetilde{\lambda}_{ui}} + \sqrt{\lambda_i})^2}$$

$$\geq (\sqrt{\widetilde{\lambda}_{uk}} + \sqrt{\lambda_k}) \sqrt{\sum_{i=1}^k (\sqrt{\widetilde{\lambda}_{ui}} - \sqrt{\lambda_i})^2}$$

$$> (1.5 - 0.1\kappa^{-1})\sqrt{\lambda_k} \|\widetilde{\mathbf{\Lambda}}_u^{\frac{1}{2}} - \mathbf{\Lambda}_k^{\frac{1}{2}}\|_F$$

$$> 0.1\kappa^{-1}(1.5 - 0.1\kappa^{-1})\lambda_k.$$

Now, we are ready to bound the square root of the second term in (D.12). In particular, we have

$$\|\mathbf{W}_k \widetilde{\mathbf{\Lambda}}_u^{\frac{1}{2}} (\widetilde{\mathbf{\Lambda}}_u - \mathbf{\Lambda}_k) \widetilde{\mathbf{Q}}_u^\top + \mathbf{W}_k \widetilde{\mathbf{\Lambda}}_u^{\frac{1}{2}} \widetilde{\mathbf{Q}}_u^\top \widetilde{\mathbf{E}}_u^\top \widetilde{\mathbf{E}}_u\|_F$$

$$\overset{①}{=} \|\widetilde{\mathbf{\Lambda}}_u^{\frac{1}{2}} [(\widetilde{\mathbf{\Lambda}}_u - \mathbf{\Lambda}_k)\widetilde{\mathbf{Q}}_u^\top + \widetilde{\mathbf{Q}}_u^\top \widetilde{\mathbf{E}}_u^\top \widetilde{\mathbf{E}}_u]\|_F$$

$$\overset{②}{\geq} \sqrt{\widetilde{\lambda}_{uk}} \|(\widetilde{\mathbf{\Lambda}}_u - \mathbf{\Lambda}_k)\widetilde{\mathbf{Q}}_u^\top + \widetilde{\mathbf{Q}}_u^\top \widetilde{\mathbf{E}}_u^\top \widetilde{\mathbf{E}}_u\|_F \qquad\qquad (\text{D.14})$$

$$\geq \sqrt{\widetilde{\lambda}_{uk}} (\|\widetilde{\mathbf{\Lambda}}_u - \mathbf{\Lambda}_k\|_F - \|\widetilde{\mathbf{E}}_u\|_F^2)$$

$$> (0.5 - 0.1\kappa^{-1})\sqrt{\lambda_k}(0.1\kappa^{-1}(1.5 - 0.1\kappa^{-1})\lambda_k - 0.01\kappa^{-2}\lambda_k)$$

$$= (0.5 - 0.1\kappa^{-1})(0.15\kappa^{-1} - 0.02\kappa^{-2})\lambda_k^{\frac{3}{2}},$$

where ① follows from $\mathbf{W}_k^\top \mathbf{W}_k = \mathbf{I}_k$, and ② follows from [11, Corollary 2].

Note that

$$(0.5 - 0.1\kappa^{-1})(0.15\kappa^{-1} - 0.02\kappa^{-2}) \geq \frac{1}{60}\kappa^{-1}$$

always holds for $\kappa \geq 1$. By combining (D.12), (D.13) and (D.14), we get

$$\|\operatorname{grad} g(\mathbf{U})\|_F > \frac{1}{60}\kappa^{-1}\lambda_k^{\frac{3}{2}}.$$

Thus, we finish the proof of second inequality in property (4).

### D.4.2 Large gradient in region $\mathcal{R}_3''$

For any $\mathbf{U} \in \mathbb{R}_*^{N \times k}$, denote $\{\sigma_i\}_{i=1}^k$ as its singular values. Then, by using the Cauchy-Schwarz inequality, we have

$$\|\mathbf{U}\|_F^2 = \sum_{i=1}^k \sigma_i^2 \leq \sqrt{k} \sqrt{\sum_{i=1}^k \sigma_i^4} = \sqrt{k}\|\mathbf{U}\mathbf{U}^\top\|_F. \qquad\qquad (\text{D.15})$$

On one hand, we have

$$\langle \operatorname{grad} g(\mathbf{U}), \mathbf{U} \rangle \leq \|\operatorname{grad} g(\mathbf{U})\|_F \|\mathbf{U}\|_F \leq k^{\frac{1}{4}} \|\operatorname{grad} g(\mathbf{U})\|_F \|\mathbf{U}\mathbf{U}^\top\|_F^{\frac{1}{2}}. \qquad (\text{D.16})$$

On the other hand, we have

$$\langle \operatorname{grad} g(\mathbf{U}), \mathbf{U} \rangle = \langle (\mathbf{U}\mathbf{U}^\top - \mathbf{X})\mathbf{U}, \mathbf{U} \rangle$$

$$= \langle \mathbf{U}\mathbf{U}^\top - \mathbf{U}^\star \mathbf{U}^{\star\top}, \mathbf{U}\mathbf{U}^\top \rangle - \langle \mathbf{W}_k^\perp \mathbf{\Lambda}_k^\perp \mathbf{W}_k^{\perp\top}, \mathbf{U}\mathbf{U}^\top \rangle$$

$$\overset{①}{\geq} \|\mathbf{U}\mathbf{U}^\top\|_F^2 - \|\mathbf{U}^\star \mathbf{U}^{\star\top}\|_F \|\mathbf{U}\mathbf{U}^\top\|_F - \|\mathbf{W}_k^\perp \mathbf{\Lambda}_k^\perp \mathbf{W}_k^{\perp\top}\|_2 \|\mathbf{U}\mathbf{U}^\top\|_*$$

$$\overset{②}{>} \frac{1}{8}\|\mathbf{U}\mathbf{U}^\top\|_F^2 - \lambda_{k+1}\sqrt{k}\|\mathbf{U}\mathbf{U}^\top\|_F, \qquad\qquad (\text{D.17})$$

$$\overset{③}{>} \frac{1}{7}\|\mathbf{U}\mathbf{U}^\top\|_F \|\mathbf{U}^\star \mathbf{U}^{\star\top}\|_F - \frac{1}{12}\lambda_k\sqrt{k}\|\mathbf{U}\mathbf{U}^\top\|_F$$

$$\geq \frac{1}{7}\sqrt{k}\lambda_k\|\mathbf{U}\mathbf{U}^\top\|_F - \frac{1}{12}\sqrt{k}\lambda_k\|\mathbf{U}\mathbf{U}^\top\|_F$$

$$= \frac{5}{84}\sqrt{k}\lambda_k\|\mathbf{U}\mathbf{U}^\top\|_F$$

where ① follows from the Matrix Hölder Inequality [12], and ② and ③ follow from $\|\mathbf{U}^\star\mathbf{U}^{\star\top}\|_F < \frac{7}{8}\|\mathbf{U}\mathbf{U}^\top\|_F$ and $\lambda_{k+1} \leq \frac{1}{12}\lambda_k$. Combining (D.16) and (D.17), we get

$$\|\operatorname{grad} g(\mathbf{U})\|_F > \frac{5}{84}k^{\frac{1}{4}}\lambda_k\|\mathbf{U}\mathbf{U}^\top\|_F^{\frac{1}{2}} > \frac{5}{84}k^{\frac{1}{4}}\lambda_k\|\mathbf{U}^\star\mathbf{U}^{\star\top}\|_F^{\frac{1}{2}} \geq \frac{5}{84}k^{\frac{1}{4}}\lambda_k\|\mathbf{U}^\star\|_2 \geq \frac{5}{84}k^{\frac{1}{4}}\lambda_k^{\frac{3}{2}},$$

where the second to last inequality follows from $\|\mathbf{U}^\star\mathbf{U}^{\star\top}\|_F \geq \|\mathbf{U}^\star\|_2^2$. Thus, we finish the proof of the third inequality in property (4).

# E   Proof of Lemma 3.2

We present the Riemannian gradient and Hessian of the empirical risk on the quotient manifold $\mathcal{M}$ as follows

$$\operatorname{grad} f(\mathbf{U}) = \mathcal{P}_{\mathbf{U}}(\nabla f(\mathbf{U})) = \mathcal{A}^*\mathcal{A}(\mathbf{U}\mathbf{U}^\top - \mathbf{X})\mathbf{U}$$
$$\operatorname{hess} f(\mathbf{U})[\mathbf{D}, \mathbf{D}] = \langle \mathcal{P}_{\mathbf{U}}(\nabla^2 f(\mathbf{U})[\mathbf{D}]), \mathbf{D}\rangle = \nabla^2 f(\mathbf{U})[\mathbf{D}, \mathbf{D}] - \langle \mathbf{U}\mathbf{\Omega}, \mathbf{D}\rangle = \nabla^2 f(\mathbf{U})[\mathbf{D}, \mathbf{D}]$$

for any $\mathbf{D} \in \mathcal{H}_{\mathbf{U}}\mathcal{M}$. Here, $\langle \mathbf{U}\mathbf{\Omega}, \mathbf{D}\rangle = \langle \mathbf{\Omega}, \mathbf{U}^\top\mathbf{D}\rangle = 0$ follows from the fact that $\mathbf{\Omega}$ is a skew-symmetric matrix and $\mathbf{D}^\top\mathbf{U} = \mathbf{U}^\top\mathbf{D}$.

Denote $\mathcal{B} : \mathbb{R}^{N\times N} \to \mathbb{R}^M$ as a linear operator with the $m$-th entry of the observation $\boldsymbol{y} = \mathcal{B}(\mathbf{X})$ as $\boldsymbol{y}_m = \langle \mathbf{B}_m, \mathbf{X}\rangle$. According to the way we construct the symmetric linear operator $\mathcal{A}$, i.e., $\mathbf{A}_m = \frac{1}{2}(\mathbf{B}_m + \mathbf{B}_m^\top)$, we have that

$$\|\mathcal{A}(\mathbf{Z})\|_2^2 = \sum_{m=1}^M \langle \mathbf{A}_m, \mathbf{Z}\rangle^2 = \sum_{m=1}^M \langle \mathbf{B}_m, \mathbf{Z}\rangle^2 = \|\mathcal{B}(\mathbf{Z})\|_2^2$$

holds for any symmetric matrix $\mathbf{Z} \in \mathbb{R}^{N\times N}$. Therefore, the constructed symmetric linear operator $\mathcal{A}$ satisfies the RIP condition (3.4) as long as the linear operator $\mathcal{B}$ satisfies the RIP condition (3.4).

Since the linear operator $\mathcal{A}$ satisfies the RIP condition (3.4) for any matrix $\mathbf{Z} \in \mathbb{R}^{N\times N}$ with rank at most $r + k$, we have

$$\|\mathcal{A}^*\mathcal{A}(\mathbf{Z}) - \mathbf{Z}\|_F \leq \delta_{r+k}\|\mathbf{Z}\|_F. \tag{E.1}$$

To set the radius of the ball $\mathcal{B}(l) = \{\mathbf{U} \in \mathbb{R}_*^{N\times k} : \|\mathbf{U}\mathbf{U}^\top\|_F \leq l\}$, we first bound $\|\operatorname{grad} f(\mathbf{U})\|_F$ in $\mathcal{R}_3''$. On one hand, we have

$$\langle \operatorname{grad} f(\mathbf{U}), \mathbf{U}\rangle \leq \|\operatorname{grad} f(\mathbf{U})\|_F\|\mathbf{U}\|_F \leq k^{\frac{1}{4}}\|\operatorname{grad} f(\mathbf{U})\|_F\|\mathbf{U}\mathbf{U}^\top\|_F^{\frac{1}{2}},$$

which follows from the Matrix Hölder Inequality [12] and (D.15). On the other hand, we have

$$\langle \operatorname{grad} f(\mathbf{U}), \mathbf{U}\rangle$$
$$= \|\mathcal{A}(\mathbf{U}\mathbf{U}^\top)\|_2^2 - \langle \mathcal{A}(\mathbf{U}^\star\mathbf{U}^{\star\top}), \mathcal{A}(\mathbf{U}\mathbf{U}^\top)\rangle - \langle \mathcal{A}(\mathbf{W}_k^\perp\mathbf{\Lambda}_k^\perp\mathbf{W}_k^{\perp\top}), \mathcal{A}(\mathbf{U}\mathbf{U}^\top)\rangle$$
$$\overset{①}{\geq} \|\mathcal{A}(\mathbf{U}\mathbf{U}^\top)\|_2^2 - \|\mathcal{A}(\mathbf{U}^\star\mathbf{U}^{\star\top})\|_2\|\mathcal{A}(\mathbf{U}\mathbf{U}^\top)\|_2 - \|\mathcal{A}(\mathbf{W}_k^\perp\mathbf{\Lambda}_k^\perp\mathbf{W}_k^{\perp\top})\|_2\|\mathcal{A}(\mathbf{U}\mathbf{U}^\top)\|_2$$
$$\overset{②}{\geq} (1-\delta_{r+k})\|\mathbf{U}\mathbf{U}^\top\|_F^2 - (1+\delta_{r+k})\|\mathbf{U}^\star\mathbf{U}^{\star\top}\|_F\|\mathbf{U}\mathbf{U}^\top\|_F - (1+\delta_{r+k})\|\mathbf{W}_k^\perp\mathbf{\Lambda}_k^\perp\mathbf{W}_k^{\perp\top}\|_F\|\mathbf{U}\mathbf{U}^\top\|_F$$
$$\overset{③}{\geq} \frac{1}{7}(1 - 15\delta_{r+k})\|\mathbf{U}^\star\mathbf{U}^{\star\top}\|_F\|\mathbf{U}\mathbf{U}^\top\|_F - (1 + \delta_{r+k})\|\mathbf{W}_k^\perp\mathbf{\Lambda}_k^\perp\mathbf{W}_k^{\perp\top}\|_F\|\mathbf{U}\mathbf{U}^\top\|_F$$
$$\overset{④}{\geq} (\frac{5}{84} - \frac{15}{7}\delta_{r+k})\sqrt{k}\lambda_k\|\mathbf{U}\mathbf{U}^\top\|_F.$$

Here, ① follows from the Hölder's Inequality. ② follows from the RIP condition in (3.4), $\operatorname{rank}(\mathbf{U}\mathbf{U}^\top) = \operatorname{rank}(\mathbf{U}^\star\mathbf{U}^{\star\top}) = k \leq r+k$ and $\operatorname{rank}(\mathbf{W}_k^\perp\mathbf{\Lambda}_k^\perp\mathbf{W}_k^{\perp\top}) = r-k \leq k \leq r+k$. ③ follows from $\|\mathbf{U}\mathbf{U}^\top\|_F \geq \frac{8}{7}\|\mathbf{U}^\star\mathbf{U}^{\star\top}\|_F$. ④ follows from $\|\mathbf{U}^\star\mathbf{U}^{\star\top}\|_F = \|\mathbf{\Lambda}_k\|_F = \sqrt{\sum_{i=1}^k \lambda_i^2} \geq$

$\sqrt{k}\lambda_k$ and $\|\mathbf{W}_k^\perp\mathbf{\Lambda}_k^\perp\mathbf{W}_k^{\perp\top}\|_F \leq \sqrt{k}\|\mathbf{W}_k^\perp\mathbf{\Lambda}_k^\perp\mathbf{W}_k^{\perp\top}\|_2 = \sqrt{k}\lambda_{k+1} \leq \frac{1}{12}\sqrt{k}\lambda_k$. It follows that

$$\|\operatorname{grad} f(\mathbf{U})\|_F \geq k^{-\frac{1}{4}}\|\mathbf{U}\mathbf{U}^\top\|_F^{-\frac{1}{2}}\langle \operatorname{grad} f(\mathbf{U}), \mathbf{U}\rangle$$
$$\geq (\frac{5}{84} - \frac{15}{7}\delta_{r+k})k^{\frac{1}{4}}\lambda_k\|\mathbf{U}\mathbf{U}^\top\|_F^{\frac{1}{2}}$$
$$\geq (\frac{5}{84} - \frac{15}{7}\delta_{r+k})k^{\frac{1}{4}}\lambda_k\|\mathbf{U}^\star\mathbf{U}^{\star\top}\|_F^{\frac{1}{2}}$$
$$\geq (\frac{5}{84} - \frac{15}{7}\delta_{r+k})k^{\frac{1}{4}}\lambda_k\|\mathbf{U}^\star\|_2$$
$$\geq (\frac{5}{84} - \frac{15}{7}\delta_{r+k})k^{\frac{1}{4}}\lambda_k^{\frac{3}{2}}.$$

Then, we can conclude that $\|\operatorname{grad} f(\mathbf{U})\|_F \geq (\frac{5}{84} - \frac{15}{7}\delta_{r+k})k^{\frac{1}{4}}\lambda_k^{\frac{3}{2}}$ holds when $\|\mathbf{U}\mathbf{U}^\top\|_F \geq \frac{8}{7}\|\mathbf{U}^\star\mathbf{U}^{\star\top}\|_F$. Therefore, we can set the radius of $\mathcal{B}(l) = \{\mathbf{U} \in \mathbb{R}_*^{N\times k} : \|\mathbf{U}\mathbf{U}^\top\|_F \leq l\}$ as

$$l = \frac{8}{7}\|\mathbf{U}^\star\mathbf{U}^{\star\top}\|_F.$$

Inside the ball $\mathcal{B}(l)$, we then have

$$\|\operatorname{grad} f(\mathbf{U}) - \operatorname{grad} g(\mathbf{U})\|_F = \|[\mathcal{A}^*\mathcal{A}(\mathbf{U}\mathbf{U}^\top - \mathbf{X}) - (\mathbf{U}\mathbf{U}^\top - \mathbf{X})]\mathbf{U}\|_F$$
$$\leq \|\mathcal{A}^*\mathcal{A}(\mathbf{U}\mathbf{U}^\top - \mathbf{X}) - (\mathbf{U}\mathbf{U}^\top - \mathbf{X})\|_F\|\mathbf{U}\|_F$$
$$\leq \delta_{r+k}\|\mathbf{U}\mathbf{U}^\top - \mathbf{X}\|_F k^{\frac{1}{4}}\sqrt{l}$$
$$\leq \delta_{r+k}(l + \|\mathbf{X}\|_F)k^{\frac{1}{4}}\sqrt{l},$$

which implies that

$$\|\operatorname{grad} f(\mathbf{U}) - \operatorname{grad} g(\mathbf{U})\|_F \leq \frac{\epsilon}{2}$$

if $\delta_{r+k} \leq \frac{\epsilon}{2(l+\|\mathbf{X}\|_F)k^{\frac{1}{4}}\sqrt{l}}$. As a result, if the linear operator $\mathcal{A}$ satisfies the RIP condition (3.4) with

$$\delta_{r+k} \leq \min\left\{\frac{\epsilon}{2\sqrt{\frac{8}{7}}k^{\frac{1}{4}}(\frac{8}{7}\|\mathbf{U}^\star\mathbf{U}^{\star\top}\|_F + \|\mathbf{X}\|_F)\|\mathbf{U}^\star\mathbf{U}^{\star\top}\|_F^{\frac{1}{2}}}, \frac{1}{36}\right\},$$

the Assumption 2.2 is verified. Here, the term $\frac{1}{36}$ comes from the requirement that $\frac{15}{7}\delta_{r+k} < \frac{5}{84}$.

To verify Assumption 2.3, it is enough to show that

$$|\operatorname{hess} f(\mathbf{U})[\mathbf{D}, \mathbf{D}] - \operatorname{hess} g(\mathbf{U})[\mathbf{D}, \mathbf{D}]| \leq \frac{\eta}{2}$$

holds for any $\mathbf{D} \in \mathcal{H}_\mathbf{U}\mathcal{M}$ and $\|\mathbf{D}\|_F = 1$. Note that

$$|\operatorname{hess} f(\mathbf{U})[\mathbf{D}, \mathbf{D}] - \operatorname{hess} g(\mathbf{U})[\mathbf{D}, \mathbf{D}]|$$
$$= \left|\frac{1}{2}\|\mathcal{A}(\mathbf{U}\mathbf{D}^\top + \mathbf{D}\mathbf{U}^\top)\|_2^2 - \frac{1}{2}\|\mathbf{U}\mathbf{D}^\top + \mathbf{D}\mathbf{U}^\top\|_F^2 + \langle\mathcal{A}^*\mathcal{A}(\mathbf{U}\mathbf{U}^\top - \mathbf{X}), \mathbf{D}\mathbf{D}^\top\rangle - \langle\mathbf{U}\mathbf{U}^\top - \mathbf{X}, \mathbf{D}\mathbf{D}^\top\rangle\right|$$
$$\leq \frac{1}{2}\left|\|\mathcal{A}(\mathbf{U}\mathbf{D}^\top + \mathbf{D}\mathbf{U}^\top)\|_2^2 - \|\mathbf{U}\mathbf{D}^\top + \mathbf{D}\mathbf{U}^\top\|_F^2\right| + \left|\langle\mathcal{A}^*\mathcal{A}(\mathbf{U}\mathbf{U}^\top - \mathbf{X}) - (\mathbf{U}\mathbf{U}^\top - \mathbf{X}), \mathbf{D}\mathbf{D}^\top\rangle\right|$$
$$\leq 2\delta_{r+k}\sqrt{k}l + \delta_{r+k}(l + \|\mathbf{X}\|_F) = \delta_{r+k}(2\sqrt{k}l + l + \|\mathbf{X}\|_F),$$

where the last inequality follows from

$$\left|\|\mathcal{A}(\mathbf{U}\mathbf{D}^\top + \mathbf{D}\mathbf{U}^\top)\|_2^2 - \|\mathbf{U}\mathbf{D}^\top + \mathbf{D}\mathbf{U}^\top\|_F^2\right|$$
$$\leq \delta_{r+k}\|\mathbf{U}\mathbf{D}^\top + \mathbf{D}\mathbf{U}^\top\|_F^2 \leq 4\delta_{r+k}\|\mathbf{U}\|_F^2 \leq 4\delta_{r+k}\sqrt{k}\|\mathbf{U}\mathbf{U}^\top\|_F \leq 4\delta_{r+k}\sqrt{k}l$$

and

$$\left|\langle\mathcal{A}^*\mathcal{A}(\mathbf{U}\mathbf{U}^\top - \mathbf{X}) - (\mathbf{U}\mathbf{U}^\top - \mathbf{X}), \mathbf{D}\mathbf{D}^\top\rangle\right|$$
$$\leq \|\mathcal{A}^*\mathcal{A}(\mathbf{U}\mathbf{U}^\top - \mathbf{X}) - (\mathbf{U}\mathbf{U}^\top - \mathbf{X})\|_F\|\mathbf{D}\mathbf{D}^\top\|_F$$
$$\leq \delta_{r+k}(l + \|\mathbf{X}\|_F)$$

by using the assumption that the linear operator $\mathcal{A}$ satisfies the RIP condition (3.4) and the fact that $\mathbf{U}\mathbf{D}^\top + \mathbf{D}\mathbf{U}^\top$ has rank at most $2k$ with $2k \le r + k$. Therefore, we can now conclude that Assumption 2.3 is verified as long as the linear operator $\mathcal{A}$ satisfies the RIP condition (3.4) with

$$\delta_{r+k} \le \frac{\eta}{2(\frac{16}{7}\sqrt{k}\|\mathbf{U}^\star\mathbf{U}^{\star\top}\|_F + \frac{8}{7}\|\mathbf{U}^\star\mathbf{U}^{\star\top}\|_F + \|\mathbf{X}\|_F)}.$$

# F  Proof of Lemma 3.3

We first consider the critical point $\boldsymbol{x} = \boldsymbol{0}$ and its neighborhood $\mathcal{R}_1$. Note that

$$\nabla^2 g(\boldsymbol{0}) = -4\boldsymbol{x}^\star\boldsymbol{x}^{\star\top} - 2\|\boldsymbol{x}^\star\|_2^2\mathbf{I}_N,$$

whose minimal eigenvalue and corresponding eigenvector are given as

$$\lambda_{\min}(\nabla^2 g(\boldsymbol{0})) = -6\|\boldsymbol{x}^\star\|_2^2 < 0,$$

$$\boldsymbol{v}_{min}(\boldsymbol{0}) = \frac{\boldsymbol{x}^\star}{\|\boldsymbol{x}^\star\|_2}.$$

Therefore, $\boldsymbol{x} = \boldsymbol{0}$ is a strict saddle point. For any $\boldsymbol{x} \in \mathcal{R}_1$, we have $\|\boldsymbol{x}\|_2 < \frac{1}{2}\|\boldsymbol{x}^\star\|_2$. Denote $\boldsymbol{v}_{\min}(\boldsymbol{x})$ as the eigenvector of $\nabla^2 g(\boldsymbol{x})$ corresponding to the smallest eigenvalue $\lambda_{\min}(\nabla^2 g(\boldsymbol{x}))$. It follows that

$$\lambda_{\min}(\nabla^2 g(\boldsymbol{x})) = \boldsymbol{v}_{\min}(\boldsymbol{x})^\top \nabla^2 g(\boldsymbol{x}) \boldsymbol{v}_{\min}(\boldsymbol{x}) \le \boldsymbol{v}_{\min}(\boldsymbol{0})^\top \nabla^2 g(\boldsymbol{x}) \boldsymbol{v}_{\min}(\boldsymbol{0})$$

$$\overset{①}{=} 12\frac{1}{\|\boldsymbol{x}^\star\|_2^2}(\boldsymbol{x}^\top\boldsymbol{x}^\star)^2 + 6\|\boldsymbol{x}\|_2^2 - 6\|\boldsymbol{x}^\star\|_2^2$$

$$\overset{②}{\le} 18\|\boldsymbol{x}\|_2^2 - 6\|\boldsymbol{x}^\star\|_2^2 \overset{③}{\le} -\frac{3}{2}\|\boldsymbol{x}^\star\|_2^2,$$

where ① follows by plugging $\boldsymbol{v}_{min}(\boldsymbol{0}) = \frac{\boldsymbol{x}^\star}{\|\boldsymbol{x}^\star\|_2}$ and $\nabla^2 g(\boldsymbol{x}) = 12\boldsymbol{x}\boldsymbol{x}^\top - 4\boldsymbol{x}^\star\boldsymbol{x}^{\star\top} + 6\|\boldsymbol{x}\|_2^2\mathbf{I}_N - 2\|\boldsymbol{x}^\star\|_2^2\mathbf{I}_N$. ② follows Cauchy-Schwarz inequality. ③ follows from $\|\boldsymbol{x}\|_2 \le \frac{1}{2}\|\boldsymbol{x}^\star\|_2$.

Next, we consider the critical point $\boldsymbol{x} = \boldsymbol{x}^\star$ and its neighborhood. The argument for another critical point $\boldsymbol{x} = -\boldsymbol{x}^\star$ is similar so we omit the proof here. Note that

$$\nabla^2 g(\boldsymbol{x}^\star) = 8\boldsymbol{x}^\star\boldsymbol{x}^{\star\top} + 4\|\boldsymbol{x}^\star\|_2^2\mathbf{I}_N,$$

whose minimal eigenvalue is

$$\lambda_{\min}(\nabla^2 g(\boldsymbol{x}^\star)) = 4\|\boldsymbol{x}^\star\|_2^2 > 0$$

with the corresponding eigenvector satisfying $\boldsymbol{v}_{\min}(\boldsymbol{x}^\star)^\top\boldsymbol{x}^\star = 0$. Therefore, $\boldsymbol{x} = \boldsymbol{x}^\star$ is a local minimum of $g(\boldsymbol{x})$. Moreover, $g(\boldsymbol{x}^\star) = 0 = \min_{\boldsymbol{x}} g(\boldsymbol{x})$ further implies that $\boldsymbol{x} = \boldsymbol{x}^\star$ is a global minimum. For any $\boldsymbol{x} \in \mathcal{R}_2$, we have $\|\boldsymbol{x} - \boldsymbol{x}^\star\|_2 \le \frac{1}{10}\|\boldsymbol{x}^\star\|_2$. Denote $\boldsymbol{v}_{\min}(\boldsymbol{x})$ as the eigenvector of $\nabla^2 g(\boldsymbol{x})$ corresponding to the smallest eigenvalue $\lambda_{\min}(\nabla^2 g(\boldsymbol{x}))$. It follows that

$$\lambda_{\min}(\nabla^2 g(\boldsymbol{x})) = \boldsymbol{v}_{\min}(\boldsymbol{x})^\top \nabla^2 g(\boldsymbol{x}) \boldsymbol{v}_{\min}(\boldsymbol{x})$$

$$= \boldsymbol{v}_{\min}(\boldsymbol{x})^\top \nabla^2 g(\boldsymbol{x}^\star) \boldsymbol{v}_{\min}(\boldsymbol{x}) - \left(\boldsymbol{v}_{\min}(\boldsymbol{x})^\top \nabla^2 g(\boldsymbol{x}^\star) \boldsymbol{v}_{\min}(\boldsymbol{x}) - \boldsymbol{v}_{\min}(\boldsymbol{x})^\top \nabla^2 g(\boldsymbol{x}) \boldsymbol{v}_{\min}(\boldsymbol{x})\right)$$

$$\ge \boldsymbol{v}_{\min}(\boldsymbol{x})^\top \nabla^2 g(\boldsymbol{x}^\star) \boldsymbol{v}_{\min}(\boldsymbol{x}) - \left|\boldsymbol{v}_{\min}(\boldsymbol{x})^\top \left(\nabla^2 g(\boldsymbol{x}) - \nabla^2 g(\boldsymbol{x}^\star)\right) \boldsymbol{v}_{\min}(\boldsymbol{x})\right|.$$

Then, we bound the two terms on the right hand side in sequence. For the first term, we have

$$\boldsymbol{v}_{\min}(\boldsymbol{x})^\top \nabla^2 g(\boldsymbol{x}^\star) \boldsymbol{v}_{\min}(\boldsymbol{x}) = 8(\boldsymbol{x}^{\star\top}\boldsymbol{v}_{\min}(\boldsymbol{x}))^2 + 4\|\boldsymbol{x}^\star\|_2^2 \ge 4\|\boldsymbol{x}^\star\|_2^2.$$

Define $\boldsymbol{e} = \boldsymbol{x} - \boldsymbol{x}^\star$. For the second term, we have

$$\left|\boldsymbol{v}_{\min}(\boldsymbol{x})^\top \left(\nabla^2 g(\boldsymbol{x}) - \nabla^2 g(\boldsymbol{x}^\star)\right) \boldsymbol{v}_{\min}(\boldsymbol{x})\right|$$

$$= \left|24\boldsymbol{v}_{\min}(\boldsymbol{x})^\top \boldsymbol{x}^\star \boldsymbol{e}^\top \boldsymbol{v}_{\min}(\boldsymbol{x}) + 12(\boldsymbol{e}^\top \boldsymbol{v}_{\min}(\boldsymbol{x}))^2 + 12\boldsymbol{e}^\top \boldsymbol{x}^\star + 6\|\boldsymbol{e}\|_2^2\right|$$

$$\le 36\|\boldsymbol{x}^\star\|_2\|\boldsymbol{e}\|_2 + 18\|\boldsymbol{e}\|_2^2$$

$$\le 3.78\|\boldsymbol{x}^\star\|_2^2,$$

where the last two inequalities follow from the Cauchy-Schwarz inequality and $\|e\|_2 \leq \frac{1}{10}\|x^\star\|_2$. Therefore, we have

$$\lambda_{\min}(\nabla^2 g(x)) \geq 4\|x^\star\|_2^2 - 3.78\|x^\star\|_2^2 = 0.22\|x^\star\|_2^2.$$

Then, we consider the critical points $x = \frac{1}{\sqrt{3}}\|x^\star\|_2 w$, with $w^\top x^\star = 0$, $\|w\|_2 = 1$ and its neighborhood $\mathcal{R}_3$. The argument for the other critical point $x = -\frac{1}{\sqrt{3}}\|x^\star\|_2 w$ is similar so we omit the proof here. Note that

$$\nabla^2 g(\frac{1}{\sqrt{3}}\|x^\star\|_2 w) = 4\|x^\star\|_2^2 ww^\top - 4x^\star x^{\star\top},$$

whose minimal eigenvalue and corresponding eigenvector are given as

$$\lambda_{\min}(\nabla^2 g(\frac{1}{\sqrt{3}}\|x^\star\|_2 w)) = -4\|x^\star\|_2^2 < 0,$$

$$v_{min}(\mathbf{0}) = \frac{x^\star}{\|x^\star\|_2}.$$

Therefore, $x = \frac{1}{\sqrt{3}}\|x^\star\|_2 w$ with $w^\top x^\star = 0$, $\|w\|_2 = 1$ are strict saddle points. For any $x \in \mathcal{R}_3$, we have $\|x - \frac{1}{\sqrt{3}}\|x^\star\|_2 w\|_2 \leq \frac{1}{5}\|x^\star\|_2$. Denote $v_{\min}(x)$ as the eigenvector of $\nabla^2 g(x)$ corresponding to the smallest eigenvalue $\lambda_{\min}(\nabla^2 g(x))$. It follows that

$$\lambda_{\min}(\nabla^2 g(x)) = v_{\min}(x)^\top \nabla^2 g(x) v_{\min}(x) \leq v_{\min}(\mathbf{0})^\top \nabla^2 g(x) v_{\min}(\mathbf{0})$$

$$\overset{①}{=} 12\frac{1}{\|x^\star\|_2^2}(x^\top x^\star)^2 + 6\|x\|_2^2 - 6\|x^\star\|_2^2 \overset{②}{\leq} 18\|x\|_2^2 - 6\|x^\star\|_2^2$$

$$= 18\left\|x - \frac{1}{\sqrt{3}}\|x^\star\|_2 w + \frac{1}{\sqrt{3}}\|x^\star\|_2 w\right\|_2^2 - 6\|x^\star\|_2^2$$

$$\leq 18\left\|x - \frac{1}{\sqrt{3}}\|x^\star\|_2 w\right\|_2^2 + \frac{1}{3}\|x^\star\|_2^2 + \frac{36}{\sqrt{3}}\left\|x - \frac{1}{\sqrt{3}}\|x^\star\|_2 w\right\|_2 \|x^\star\|_2 - 6\|x^\star\|_2^2$$

$$\overset{③}{\leq} -0.78\|x^\star\|_2^2,$$

where ① follows by plugging $v_{min}(\mathbf{0}) = \frac{x^\star}{\|x^\star\|_2}$ and $\nabla^2 g(x) = 12xx^\top - 4x^\star x^{\star\top} + 6\|x\|_2^2 \mathbf{I}_N - 2\|x^\star\|_2^2 \mathbf{I}_N$. ② follows from the Cauchy-Schwarz inequality. ③ follows from $\|x - \frac{1}{\sqrt{3}}\|x^\star\|_2 w\|_2 \leq \frac{1}{5}\|x^\star\|_2$.

Finally, we show that the gradient $\nabla g(x)$ has a sufficiently large norm when $x \in \mathcal{R}_4$. Let $x = \alpha x^\star + \beta\|x^\star\|_2 w$ with $\alpha, \beta \in \mathbb{R}$, $w^\top x^\star = 0$, and $\|w\|_2 = 1$. Then, $\|x\|_2 > \frac{1}{2}\|x^\star\|_2$, $\min_{\gamma \in \{-1,1\}}\|x - \gamma x^\star\|_2 > \frac{1}{10}\|x^\star\|_2$ and $\min_{\gamma \in \{1,-1\}}\left\|x - \gamma\frac{1}{\sqrt{3}}\|x^\star\|_2 w\right\|_2 > \frac{1}{5}\|x^\star\|_2$ are equivalent to

$$\begin{cases} \alpha^2 + \beta^2 > \frac{1}{4}, \\ \min_{\gamma \in \{-1,1\}}(\alpha - \gamma)^2 + \beta^2 > \frac{1}{100}, \\ \min_{\gamma \in \{-1,1\}}\alpha^2 + \left(\beta - \frac{1}{\sqrt{3}}\gamma\right)^2 > \frac{1}{25}. \end{cases}$$

Note that

$$\|\nabla g(x)\|_2^2 = \left\|6\|x\|_2^2 x - 2\|x^\star\|_2^2 x - 4(x^{\star\top}x)x^\star\right\|_2^2$$

$$= 4\left(9\alpha^2(\alpha^2 + \beta^2 - 1)^2 + \beta^2(3\alpha^2 + 3\beta^2 - 1)^2\right)\|x^\star\|_2^6$$

$$> 0.1571\|x^\star\|_2^6.$$

Then, we have

$$\|\nabla g(x)\|_2 > 0.3963\|x^\star\|_2^3.$$

# G   Proof of Lemma 3.4

The gradient and Hessian of the empirical risk (1.1) are given as

$$\nabla f(\boldsymbol{x}) = \frac{2}{M} \sum_{m=1}^{M} (\boldsymbol{a}_m \langle \boldsymbol{a}_m, \boldsymbol{x} \rangle^3 - \boldsymbol{a}_m \langle \boldsymbol{a}_m, \boldsymbol{x} \rangle \langle \boldsymbol{a}_m, \boldsymbol{x}^\star \rangle^2),$$

$$\nabla^2 f(\boldsymbol{x}) = \frac{2}{M} \sum_{m=1}^{M} (3\boldsymbol{a}_m \boldsymbol{a}_m^\top \langle \boldsymbol{a}_m, \boldsymbol{x} \rangle^2 - \boldsymbol{a}_m \boldsymbol{a}_m^\top \langle \boldsymbol{a}_m, \boldsymbol{x}^\star \rangle^2).$$

Observe that

$$\|\nabla f(\boldsymbol{x}) - \nabla g(\boldsymbol{x})\|_2$$

$$= 2 \left\| \frac{1}{M} \sum_{m=1}^{M} \boldsymbol{a}_m \langle \boldsymbol{a}_m, \boldsymbol{x} \rangle^3 - 3\|\boldsymbol{x}\|_2^2 \boldsymbol{x} - \frac{1}{M} \sum_{m=1}^{M} \boldsymbol{a}_m \langle \boldsymbol{a}_m, \boldsymbol{x} \rangle \langle \boldsymbol{a}_m, \boldsymbol{x}^\star \rangle^2 + \|\boldsymbol{x}^\star\|_2^2 \boldsymbol{x} + 2(\boldsymbol{x}^{\star^\top} \boldsymbol{x}) \boldsymbol{x}^\star \right\|_2$$

$$\leq 2 \left\| \frac{1}{M} \sum_{m=1}^{M} \boldsymbol{a}_m \langle \boldsymbol{a}_m, \boldsymbol{x} \rangle^3 - 3\|\boldsymbol{x}\|_2^2 \boldsymbol{x} \right\|_2 + 2 \left\| \frac{1}{M} \sum_{m=1}^{M} \boldsymbol{a}_m \langle \boldsymbol{a}_m, \boldsymbol{x} \rangle \langle \boldsymbol{a}_m, \boldsymbol{x}^\star \rangle^2 - \|\boldsymbol{x}^\star\|_2^2 \boldsymbol{x} - 2(\boldsymbol{x}^{\star^\top} \boldsymbol{x}) \boldsymbol{x}^\star \right\|_2.$$

To bound the above two terms, we need the following lemma, which is a direct result from [15, Claim 5] by setting $\mathbf{A} = \mathbf{I}_N$ and $k = d = N$.

**Lemma G.1.** *Suppose $\boldsymbol{a}_m \in \mathbb{R}^N$ is a Gaussian random vector with entries satisfying $\mathcal{N}(0, 1)$. Denote $\boldsymbol{a}_m^{\otimes 4} = \boldsymbol{a}_m \otimes \boldsymbol{a}_m \otimes \boldsymbol{a}_m \otimes \boldsymbol{a}_m \in \mathbb{R}^{N \times N \times N \times N}$ as a fourth order tensor. Then, we have*

$$\left\| \frac{1}{M} \sum_{m=1}^{M} \left( \boldsymbol{a}_m^{\otimes 4} - \mathbb{E} \boldsymbol{a}_m^{\otimes 4} \right) \right\|_2 \leq \widetilde{\mathcal{O}} \left( \frac{N^2}{M} + \sqrt{\frac{N}{M}} \right) \triangleq h(N, M)$$

*holds with probability at least $1 - e^{-CN \log(M)}$.*

For the first term, we have

$$2 \left\| \frac{1}{M} \sum_{m=1}^{M} \boldsymbol{a}_m \langle \boldsymbol{a}_m, \boldsymbol{x} \rangle^3 - 3\|\boldsymbol{x}\|_2^2 \boldsymbol{x} \right\|_2$$

$$= 2 \left\| \frac{1}{M} \sum_{m=1}^{M} \left( \boldsymbol{a}_m^{\otimes 4} - \mathbb{E} \boldsymbol{a}_m^{\otimes 4} \right) \times_1 \boldsymbol{x} \times_2 \boldsymbol{x} \times_3 \boldsymbol{x} \right\|_2$$

$$\leq 2 \left\| \frac{1}{M} \sum_{m=1}^{M} \left( \boldsymbol{a}_m^{\otimes 4} - \mathbb{E} \boldsymbol{a}_m^{\otimes 4} \right) \right\|_2 \|\boldsymbol{x}\|_2^3$$

$$\leq 2h(N, M)l^3,$$

where the last inequality follows from Lemma G.1 and $\|\boldsymbol{x}\|_2 \leq l$.

For the second term, we have

$$2 \left\| \frac{1}{M} \sum_{m=1}^{M} \boldsymbol{a}_m \langle \boldsymbol{a}_m, \boldsymbol{x} \rangle \langle \boldsymbol{a}_m, \boldsymbol{x}^\star \rangle^2 - \|\boldsymbol{x}^\star\|_2^2 \boldsymbol{x} - 2(\boldsymbol{x}^{\star^\top} \boldsymbol{x}) \boldsymbol{x}^\star \right\|_2$$

$$= 2 \left\| \frac{1}{M} \sum_{m=1}^{M} \left( \boldsymbol{a}_m^{\otimes 4} - \mathbb{E} \boldsymbol{a}_m^{\otimes 4} \right) \times_1 \boldsymbol{x} \times_2 \boldsymbol{x}^\star \times_3 \boldsymbol{x}^\star \right\|_2$$

$$\leq 2 \left\| \frac{1}{M} \sum_{m=1}^{M} \left( \boldsymbol{a}_m^{\otimes 4} - \mathbb{E} \boldsymbol{a}_m^{\otimes 4} \right) \right\|_2 \|\boldsymbol{x}\|_2 \|\boldsymbol{x}^\star\|_2^2$$

$$\leq 2h(N, M)l\|\boldsymbol{x}^\star\|_2^2,$$

where the last inequality follows from Lemma G.1 and $\|\boldsymbol{x}\|_2 \leq l$.

Therefore, we have that

$$\|\nabla f(\boldsymbol{x}) - \nabla g(\boldsymbol{x})\|_2 \leq 2h(N, M)l(l^2 + \|\boldsymbol{x}^\star\|_2^2) \leq \frac{\epsilon}{2}$$

holds with probability at least $1 - e^{-CN\log(M)}$ if

$$h(N, M) \leq \frac{\epsilon}{4l(l^2 + \|\boldsymbol{x}^\star\|_2^2)}. \tag{G.1}$$

As is stated in Lemma 3.3, we have shown that $\|\nabla g(\boldsymbol{x})\|_2 \geq \epsilon$ in $\mathcal{R}_4$. Set the radius of the ball $\mathcal{B}^N(l) \triangleq \{\boldsymbol{x} \in \mathbb{R}^N : \|\boldsymbol{x}\|_2 \leq l\}$ as $l = 1.1\|\boldsymbol{x}^\star\|_2$. It can be seen that the region outside the ball $\mathcal{B}^N(l)$ is a subset of $\mathcal{R}_4$. Thus, we still have $\|\nabla g(\boldsymbol{x})\|_2 \geq \epsilon$ when $\boldsymbol{x} \notin \mathcal{B}^N(l)$. Then, for any $\boldsymbol{x} \notin \mathcal{B}^N(l)$, we have that

$$\|\nabla f(\boldsymbol{x})\|_2 = \|\nabla g(\boldsymbol{x}) + (\nabla f(\boldsymbol{x}) - \nabla g(\boldsymbol{x}))\|_2$$
$$\geq \|\nabla g(\boldsymbol{x})\|_2 - \|\nabla f(\boldsymbol{x}) - \nabla g(\boldsymbol{x})\|_2 \geq \frac{\epsilon}{2}$$

holds with probability at least $1 - e^{-CN\log(M)}$. Here, we have used $\|\nabla f(\boldsymbol{x}) - \nabla g(\boldsymbol{x})\|_2 \leq \frac{\epsilon}{2}$ with high probability and $\|\nabla g(\boldsymbol{x})\|_2 \geq \epsilon$.

Since $f(\boldsymbol{x})$ has a large gradient when $\boldsymbol{x} \notin \mathcal{B}^N(l)$ with $l = 1.1\|\boldsymbol{x}^\star\|_2$, we only need to consider the geometry of $f(\boldsymbol{x})$ with $\boldsymbol{x} \in \mathcal{B}^N(l)$. Then, by plugging $l = 1.1\|\boldsymbol{x}^\star\|_2$ and $\epsilon = 0.3963\|\boldsymbol{x}^\star\|_2^3$ into (G.1), we get

$$h(N, M) \leq 0.0407.$$

Similarly, we can show that

$$\|\nabla^2 f(\boldsymbol{x}) - \nabla^2 g(\boldsymbol{x})\|_2$$
$$\leq 6\left\|\frac{1}{M}\sum_{m=1}^M \left(\boldsymbol{a}_m^{\otimes 4} - \mathbb{E}\boldsymbol{a}_m^{\otimes 4}\right) \times_1 \boldsymbol{x} \times_2 \boldsymbol{x}\right\|_2 + 2\left\|\frac{1}{M}\sum_{m=1}^M \left(\boldsymbol{a}_m^{\otimes 4} - \mathbb{E}\boldsymbol{a}_m^{\otimes 4}\right) \times_1 \boldsymbol{x}^\star \times_2 \boldsymbol{x}^\star\right\|_2$$
$$\leq 6\left\|\frac{1}{M}\sum_{m=1}^M \left(\boldsymbol{a}_m^{\otimes 4} - \mathbb{E}\boldsymbol{a}_m^{\otimes 4}\right)\right\|_2 \|\boldsymbol{x}\|_2^2 + 2\left\|\frac{1}{M}\sum_{m=1}^M \left(\boldsymbol{a}_m^{\otimes 4} - \mathbb{E}\boldsymbol{a}_m^{\otimes 4}\right)\right\|_2 \|\boldsymbol{x}^\star\|_2^2$$
$$\leq 2h(N, M)(3l^2 + \|\boldsymbol{x}^\star\|_2^2) \leq \frac{\eta}{2}$$

holds with probability at least $1 - e^{-CN\log(M)}$ if

$$h(N, M) \leq \frac{\eta}{4(3l^2 + \|\boldsymbol{x}^\star\|_2^2)}. \tag{G.2}$$

Plugging $l = 1.1\|\boldsymbol{x}^\star\|_2$ and $\eta = 0.22\|\boldsymbol{x}^\star\|_2^2$ into (G.2), we get

$$h(N, M) \leq 0.0118.$$

## Footnotes

[1]The subset $\mathcal{B}(l)$ can vary in different applications. For example, we define $\mathcal{B}(l) \triangleq \{\mathbf{U} \in \mathbb{R}_*^{N \times k} : \|\mathbf{U}\mathbf{U}^\top\|_F \leq l\}$ in matrix sensing and $\mathcal{B}(l) \triangleq \{\boldsymbol{x} \in \mathbb{R}^N : \|\boldsymbol{x}\|_2 \leq l\}$ in phase retrieval.

[3]Since the pullback is defined on a vector space, its gradient and Hessian can be computed using the regular $\nabla$ and $\nabla^2$ operators with appropriate choice of basis for $\mathcal{T}_{\boldsymbol{x}_k}\mathcal{M}$. Our notation highlights this fact.