[Reviews · NeurIPS 2019]

Reviewer 1



I have carefully read other reviewers' comments and the author feedback, and I understood that it's important to have these details in the applications section. I really appreciate the additional work done by the authors to improve the writing and contents of this paper, and these work addressed my concerns. Thus, I have raised my score. -------------------------------------------------------------------- Summary: This paper shows an interesting result which connects the landscape of the empirical risk and population risk. This result is an extension of (Song, Mei et al, 2016). The previous result establishes important connections between critical points, especially local minima, of empirical and population risk. These connections are helpful in understanding the landscape of the empirical risk and the behaviors of optimization algorithms like gradient descent. Previous work requires three assumptions: The strongly Morse property of the population risk, and the proximity for the gradient and Hessian of the population risk. The authors relaxed the first assumption and only requires the minimum eigenvalue of the Hessian is away from zero. Thus, this paper further improves the understanding of the landscape in a more general setting. They have found some examples for this new result to be applied to and done experiments to verify the theoretical results. Detailed Comments: 1. The authors spend half of the paper explaining the applications of the main result, but only use less than one page to show the main result without providing any sketch for the proof. From my point of view, this is not a very well-organized paper. The main result section is the core part of this paper, so I would expect more explanations of the main result, e.g., a proof sketch with some intuition about the significance of this result. 2. The contents in the Applications section contains too much detail. I don't think we need to look at these applications that carefully because they are just the applications of the main result. In this section, there are tons of constants like 8/7, 1.91, 0.06 and so on, which I don't think people will be interested in, and you can hide those constants using asymptotic notations or some letters. There are also details that are too technical, e.g., the division of regions, and formulas like line 196. I would suggest the authors hide this detail and make more space for the main result section. 3. The experiments only cover the case of the examples, i.e., the rank-1 matrix sensing and phase retrieval. It even doesn't cover your applications because you have proved the case for a more general matrix sensing. I think it would be better to do experiments on settings that are more general, e.g., the setting you use for the first application. 4. It would be better to have more figures in this paper to make people understand better. One figure is provided for the partition of regions in matrix sensing, so it's better to have another one for phase retrieval. Also, for the main result section, the authors can also use a figure to illustrate the assumptions or results better.

Reviewer 2



This work is an extension of a work by S. Mei et al. that looks at the landscape of empirical risk. In that work, the authors established uniform convergence of empirical risk to population risk and one to one correspondence between critical points of the two function under strongly Morse assumption. In this work, the authors relax that assumption and allow the Hessian to be be even degenerate, provided that it has a negative eigenvalue. This allows the authors to apply the theory to cases were strongly Morse assumption is violated. The authors show that the assumptions are satisfied in two examples, and in one, they show how one can use quotient manifolds to circumvent the fact that at the global min, the Hessian is degenerate, and still apply the same theorem. One minor issue with the result of the paper: can authors briefly describe how theorem 2.1 can be used to get estimation error bounds? This theorem is very strong in the sense that it provides a result for every connected compact subset of B(l). Therefore it seems reasonable to assume that getting estimation error bounds using this theorem should be doable, but the details should depend on the specific examples and underlying distributions. Establishing one to one correspondence between critical points of the two functions are definitely useful, but the ultimate goal, at least in the examples provided is to get estimation error bounds, and unfortunately it is not discussed at all. The paper is very well written and easy to follow. I did not check the details of the proofs.

Reviewer 3



The paper considers the problem of characterizing the landscape of empirical risk using the population risk. Particular attention is devoted to the case of a degenerate population risk. Their main theorem proves a close connection between stationary point and local minimizers. The paper improves on state-of-the-art in the characterization of the above mentioned landscape. Previous work could not explain the situation where the Hessian is degenerate (has zero eigenvalues) near a stationary point. The main assumption in the present paper is that the absolute value of the Hessian's minimal eigenvalue must be bounded away from zero. This includes includes (some) degenerate saddle point and strict local minimizers. It does not include degenerate minima, which is surprising (a drawback). The presentation of the paper with a separate, clearly written section on the "main result" is good, since the details are rather technical. In the section "main results", comments on the parameters eps and eta are missing. Need the conditions hold for any positive eps and eta? I think, you require only existence of eps and eta such that the assumptions are satisfied. In that case, the reader would like to see an intuitive description of the meaning of these parameters in the presented examples of matrix sensing and phase retrieval and an explanation of the proof strategy for obtaining the parameters. Is there a standard way to compute the parameters eps and eta?

[Author Response · NeurIPS 2019]

We would like to thank all three reviewers for their constructive comments and suggestions which will help us improve
the quality of this paper (ID: 1916).

**Reviewer #1:**

Q1: I would expect more explanations of the main result, e.g., a proof sketch with some
intuition about the significance of this result.
A1: Thank you for pointing this out. We will add a new paragraph or subsection after we
present Theorem 2.1 to include a proof sketch with some intuition about the significance
of our result. In this section, we will also include the estimation error bounds obtained
from Theorem 2.1 and some discussion of the parameters $\epsilon$ and $\eta$ as requested by reviewers #2 and #3.

Fig.1: estimation error bound.

Q2: In the Applications section, constants can be hidden. I would suggest the
authors hide details like the division of regions and formulas in line 196 to make
more space for the main result section.
A2: We would like to emphasize that the constants and formulas in line 196 are
critical. These constants are carefully chosen to argue that the two applications
satisfy the requisite assumptions. Similarly, we feel the partition regions are
also very important, and we did receive positive comments from reviewers
#2 and #3 concerning the paper organization. So, we are hesitant to remove
these details. However, we will be allowed a ninth content page if our paper is

Fig.2: partition regions in phase retrieval.

accepted, and we plan to devote this page to extending the discussion in the main result section. If that turns out to be
insufficient, we plan to reduce the lemmas in the application section into two informal lemmas, and move the current
formal lemmas to the supplementary material.

Q3: Do experiments on more general settings for the first application.
A3: In the revised version, we will incorporate Fig.1, which shows the distance
between the local minima of population and empirical risk in the problem of
matrix sensing with $k = 2$, $r = 3$, and $N = 8$.

Q4: Have a figure with partition regions for phase retrieval. Use a figure to
illustrate the assumptions or results.
A4: We will add Figs. 2 and 3 in the revised version. Fig. 3 is used to illustrate
the assumptions in the example of phase retrieval. In particular, we set the
parameters in Example 1.2 as $N = 1$, $x^\star = 1$, and $M = 30$. We display the
population risk and empirical risk together with their gradients and Hessians in
Fig. 3. One can see that in the small gradient region (the three parts between the
light blue vertical dashed line), $|\lambda_{\min}(\text{hess } g(\mathbf{U}))|$ (which equals the absolute

Fig.3: phase retrieval with $N = 1$.

value of the Hessian since $N = 1$) is bounded away from zero. With enough
measurements, e.g., $M = 30$, the gradients and Hessians of the empirical and population risk are close to each other. In
addition, we think our simulation figures are good illustrations of the main results in Theorem 2.1, i.e., the relationship
between critical points of empirical and population risk. We will add more details on this in the main result section.

**Reviewer #2:**

We appreciate the positive comments and will add a detailed discussion on how to obtain estimation error bounds.

**Reviewer #3:**

Q1: The main assumption in the paper does not include degenerate minima, which is surprising (a drawback).
A1: A degenerate local minimum of the population risk can correspond to a strict saddle point of the empirical risk due
to its randomness. Thus, we do not believe there exists a correspondence in the case of degenerate local minima.

Q2: Add an intuitive description of the meaning of $\epsilon$ and $\eta$ in the examples of matrix sensing and phase retrieval and an
explanation of the proof strategy for obtaining the parameters. Is there a standard way to compute $\epsilon$ and $\eta$?
A2: For phase retrieval, note that $|\lambda_{\min}(\nabla^2 g(\boldsymbol{x}))|$ and $\|\nabla g(\boldsymbol{x})\|_2$ roughly scale with $\|\boldsymbol{x}^\star\|_2^2$ and $\|\boldsymbol{x}^\star\|_2^3$ in the regions
near critical points, which implies that $\eta$ and $\epsilon$ should also scale with $\|\boldsymbol{x}^\star\|_2^2$ and $\|\boldsymbol{x}^\star\|_2^3$, respectively. For matrix
sensing, in a similar way, $|\lambda_{\min}(\text{hess } g(\mathbf{U}))|$ and $\|\text{grad } g(\mathbf{U})\|_F$ roughly scale with $\lambda_k$ and $\lambda_k^{1.5}$ in the regions near
critical points, which implies that $\eta$ and $\epsilon$ should also scale with $\lambda_k$ and $\lambda_k^{1.5}$, respectively. We will incorporate a more
detailed discussion on this in the revised version. To obtain these parameters, one can lower bound $|\lambda_{\min}(\text{hess}(g))|$ in a
small gradient region. In this way, one can adjust the size of the small gradient region to get $\epsilon$, and use the lower bound
for $|\lambda_{\min}(\text{hess}(g))|$ as $\eta$. In the case when it is not easy to directly bound $|\lambda_{\min}(\text{hess}(g))|$ in a small gradient region,
one can also first choose a region for which it is easy to find the lower bound, and then argue that the gradient has a
large norm outside of this region, as we did in this paper.

[Meta-Review · NeurIPS 2019]

This paper generalizes the result of Mei and Montanari to show that (under some assumptions) as long as the population risk has nice behavior, the empirical risk also have nice properties. The main novelty is that it removed the assumption that saddle points need to be hyperbolic - this is important for the applications.